# BOOLEAN SATISFIABILITY VIA IMITATION LEARNING

**Zewei Zhang**[1]  **Huan Liu**[1]  **Yuanhao Yu**[1]  **Jun Chen**[1]  **Xiangyu Xu**[2]*
[1]McMaster University  [2]Xi'an Jiaotong University
`{zhanz561, chenjun}@mcmaster.ca`  `{liuh127, dr.yhyu}@outlook.com`
`xuxiangyu2014@gmail.com`

## ABSTRACT

We propose ImitSAT, a branching policy for conflict-driven clause learning (CDCL) solvers based on imitation learning for the Boolean satisfiability problem (SAT). Unlike previous methods that predict instance-level signals to improve CDCL branching indirectly, or rely on reinforcement learning and insufficient CDCL information to enhance branching, ImitSAT learns from expert KeyTrace that collapses a full run into the sequence of surviving decisions. Replaying a KeyTrace on the same instance is nearly conflict-free, providing dense decision-level supervision and directly reducing propagations—the dominant contributor to wall-clock time. This prefix-conditioned supervision enables ImitSAT to reproduce high-quality branches without exploration, yielding faster convergence, stable training, and seamless integration into CDCL. Extensive experiments demonstrate that ImitSAT reduces propagation counts and runtime, outperforming state-of-the-art learned approaches. We released the source code and trained model at `https://github.com/zewei-Zhang/ImitSAT`.

## 1 INTRODUCTION

The Boolean satisfiability (SAT) problem is a cornerstone of theoretical computer science and artificial intelligence (Cook, 1971; Karp, 1972). Beyond its foundational role, SAT serves as the computational backbone of numerous applications, including formal verification, planning, and combinatorial optimization. Modern solvers for SAT are dominated by the conflict-driven clause learning (CDCL) framework (Silva & Sakallah, 1996; Biere et al., 2009), which has scaled to industrial benchmarks of immense complexity. A CDCL run interleaves branching, unit propagation, and conflict analysis. Among these components, the branching rule largely determines the search trajectory, while unit propagation often dominates runtime (Zhang & Malik, 2002; Davis et al., 2008; Moskewicz et al., 2001). As a result, more informed branching decisions can translate directly into faster solving.

Classical branching heuristics, however, are hand-crafted and limited in their adaptability. Recent work has sought to improve solver performance by integrating learning-based guidance. For example, SATformer (Shi et al., 2023) learns instance-level signals to adjust variable activities during initialization, but it exerts no influence once the branching loop begins. Graph-Q-SAT (Kurin et al., 2020) introduces an online agent within CDCL, yet it relies on reinforcement learning (RL), which requires extensive exploration and can be unstable due to sparse rewards and delayed feedback. Moreover, Graph-Q-SAT does not utilize the full CDCL execution history; each branching is conditioned only on a compact graph snapshot of the current state, so the history influences the agent only indirectly.

In contrast, we adopt imitation learning, which trains directly from expert traces. Specifically, we introduce ImitSAT, a CDCL branching learner trained on expert trace replays. Replaying an expert trace on the same instance is nearly conflict-free, eliminating redundant propagations and providing clean training targets. Since branching is inherently prefix-conditioned, we formulate it as an autoregressive sequence modeling problem, and implement ImitSAT with a Transformer-based learner (Vaswani et al., 2017). Our model captures long-context dependencies while keeping runtime costs practical through lightweight autoregressive attention (Hawthorne et al., 2022).

---

*Corresponding author

To obtain expert traces, we collapse full solver runs into sequences of surviving decisions, yielding dense, step-level supervision at every branching point. This approach enables the learner to reproduce high-quality decisions without costly exploration, resulting in faster convergence, more stable training, and natural alignment with the prefix-conditioned structure of branching. During inference, ImitSAT reads the instance together with a KeyTrace prefix and autoregressively predicts the next signed variable as the branch decision. The policy is queried under a small budget, reverting to the native heuristic when uncertain, while all other CDCL components remain unchanged, preserving completeness and robustness.

We evaluate ImitSAT on held-out random 3-SAT instances ranging from 5 to 100 variables, as well as on structured families including satisfiable and unsatisfiable instances and non-$k$-SAT formulas. Under limited query budgets, ImitSAT consistently reduces propagations relative to a standard CDCL solver and outperforms prior learning-based methods, SATformer and Graph-Q-SAT, across most settings while matching their performance elsewhere. Wall-clock measurements show that ImitSAT achieves favorable runtime performance against SOTA methods (Shi et al., 2023; Kurin et al., 2020). Remarkably, although trained solely on simple random 3-SAT, ImitSAT transfers effectively to unsatisfiable and non-$k$-SAT benchmarks without modification.

We summarize the contributions of this work as below.

- We propose ImitSAT, the first branching policy for CDCL solvers based on imitation learning. Unlike prior methods that rely on reinforcement learning, ImitSAT leverages dense, decision-level supervision from expert traces.
- We cast branching as a sequential modeling problem by collapsing solver runs into compact sequences of surviving decisions. These sequences serve as clean, conflict-free training targets and align naturally with prefix-conditioned autoregressive modeling.
- Extensive experiments demonstrate that ImitSAT yields both practical efficiency and strong generalization.

## 2 RELATED WORKS

**Neural guidance for SAT and CDCL.** Early learning approaches focused on instance-level prediction, using Graph Neural Network (GNNs) (Scarselli et al., 2008) to classify SAT or UNSAT, as seen in NeuroSAT (Selsam et al., 2019; Selsam & Bjørner, 2019) and (Cameron et al., 2020). Recent work has explored whether Transformers can learn solver behavior directly (Pan et al., 2025). In parallel with these model-based approaches, complementary efforts target data and benchmarking, including G2SAT (You et al., 2019) and G4SATBench (Li et al., 2024). Building on these foundations, a second line integrates learning inside solvers to shape specific components: for example, NeuroSelect (Liu et al., 2024) learns clause deletion policies, NeuroBack (Wang et al., 2024) improves phase initialization with GNNs, and RDC-SAT (Zhai & Ge, 2025) adopts a divide-and-conquer strategy via reinforcement learning. This approach leads to targeted enhancements within solver mechanisms. More concretely, within the CDCL branching loop, several methods exemplify this integration: NeuroSAT (Selsam & Bjørner, 2019) has been used to guide variable selection; Graph-Q-SAT (Kurin et al., 2020) trains an RL agent queried online during search based on instance information; and SATformer (Shi et al., 2023) trains a GNN Transformer model to initialize the CDCL that indirectly influences branching thereafter.

**Imitation learning for control.** Imitation learning (IL) learns policies directly from expert demonstrations, that is, sequences of states with associated actions (Osa et al., 2018; Zare et al., 2024). A simple example is behavior cloning (BC), which utilizes supervised learning to map observed situations to expert choices (Pomerleau, 1991). Building on the principles of imitation learning, the Decision Transformer (Chen et al., 2021) is similar to behavior cloning, framing reinforcement learning as sequence modeling, where an autoregressive Transformer is trained to predict the next action given a sequence rollout of returns, states, and actions. This view connects control to next-token prediction and attains competitive performance without explicit value function learning. The application of imitation learning extends beyond traditional domains. For instance, beyond robotics and games, IL has guided decision-making in exact optimization solvers. In mixed-integer linear programming, policies learned to imitate strong branching can be used within branch-and-

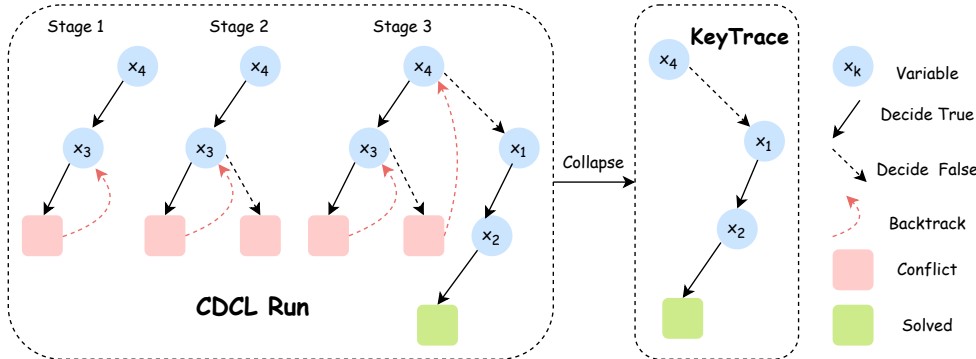

Figure 1: **From a CDCL run to a KeyTrace.** The left pane sketches one run in three stages: (1) decide $x_4 = \top$, then $x_3 = \top$, hit a conflict, backtrack to the decision on $x_3$; (2) set $x_3 = \bot$, conflict again, backtrack to $x_4$; (3) set $x_4 = \bot$, then decide $x_1 = \top$ and $x_2 = \top$ and solve. Unit assignments are omitted for clarity. Collapsing backtracks removes detours and keeps only the surviving root–to–current decisions, yielding the compact *KeyTrace* on the right, which serves as the expert for imitation. A step-by-step walkthrough is in Appendix H.

bound and achieve strong results (Gasse et al., 2019). Related work also learns branching policies within branch-and-bound (B&B) for mixed-integer programming (MIP) (Zarpellon et al., 2021; He et al., 2014; Khalil et al., 2016). While these methods are close in spirit, the underlying search dynamics differ: MIP B&B explores a monotone tree with strong-branching oracles and immediate, node-local feedback, whereas SAT CDCL performs non-monotone search with restarts, clause learning, and delayed, non-local feedback. From the imitation-learning perspective, prior MIP approaches typically learn local ranking models on MIP features, whereas we train a transformer-based autoregressive policy that imitates entire CDCL KeyTraces. Thus, our setting complements IL for MIP branching, differing both in the target solver dynamics and in the learning formulation.

Motivated by the limitations of neural guidance for SAT detailed above, and drawing inspiration from imitation learning, we propose ImitSAT. The branching policy for CDCL clones a near conflict-free KeyTrace distilled from solver runs. To achieve this, we cast branching as prefix-conditioned sequence prediction and train an autoregressive next-decision model on compact sequences of surviving decisions. This provides dense, decision-level supervision at low per-query cost. These design choices help reduce propagation and improve wall-clock time under small query budgets.

## 3 PRELIMINARIES ON SAT

**Boolean Satisfiability Problem.** The *Boolean satisfiability problem* (SAT) is the canonical decision problem in propositional logic. It asks whether a Boolean formula $F$ over variables $x_1, \ldots, x_n$ can be made true by assigning each variable a value in $\{\top, \bot\}$, where $\top$ denotes *true* and $\bot$ denotes *false*. A *literal* is either a variable $x_i$ or its negation $\neg x_i$. A *clause* is a disjunction of literals,

$$C_i = (\ell_{i,1} \vee \cdots \vee \ell_{i,k_i}), \tag{1}$$

and a formula is in *conjunctive normal form* (CNF) (Plaisted & Greenbaum, 1986) if it is a conjunction of clauses:

$$F = C_1 \wedge C_2 \wedge \cdots \wedge C_m. \tag{2}$$

An *assignment* is a mapping $\sigma : \{x_1, \ldots, x_n\} \to \{\top, \bot\}$. It satisfies a clause if at least one literal in the clause evaluates to $\top$ under $\sigma$, and it satisfies the entire formula $F$ if every clause is satisfied. We say that $F$ is *satisfiable* (SAT) if such an assignment exists, and *unsatisfiable* (UNSAT) otherwise.

A special case is the *k-SAT problem*, where each clause contains exactly $k$ literals. The case $k = 3$ (3-SAT) is of particular importance: it is NP-complete (Karp, 1972) and widely used in theoretical analysis and empirical benchmarks. For example, consider the CNF formula

$$F = (x_1 \vee \neg x_3 \vee x_4) \wedge (\neg x_1 \vee x_2 \vee x_3) \wedge (\neg x_2 \vee \neg x_3 \vee \neg x_4). \tag{3}$$

This formula is SAT, since the assignment $x_1 = \top$, $x_2 = \top$, $x_3 = \bot$, $x_4 = \bot$ makes every clause true. If no assignment satisfies all clauses, the instance would be UNSAT.

In practice, SAT formulas are often represented in the standard *DIMACS CNF format* (Johnson & Trick, 1996). In this encoding, each clause is written as a sequence of nonzero integers terminated by 0, where the integer $i$ denotes variable $x_i$ and $-i$ denotes its negation $\neg x_i$. For the formula in Equation 3, the DIMACS form is:

$$F_{\mathrm{DIMACS}} = \texttt{1 -3 4 0  -1 2 3 0  -2 -3 -4 0}. \tag{4}$$

This compact numeric encoding is convenient for algorithmic solvers and sequence-based models.

**Conflict-driven Clause Learning.**  Conflict-driven clause learning (CDCL) is the dominant algorithmic framework for practical SAT solvers (Silva & Sakallah, 1996; Biere et al., 2009). A CDCL solver incrementally explores a branching search tree through three fundamental operations: *decision* (D), where the solver assigns a value to a chosen literal and thus extends the current partial assignment; *unit propagation* (A), also known as *Boolean Constraint Propagation* (BCP) (Moskewicz et al., 2001), where implied literals are deduced from unit clauses; and *backtracking* (BT), where a detected conflict triggers clause learning and a non-chronological jump to an earlier decision level.

At each decision level, the solver selects a decision literal, after which propagation infers additional assignments, possibly none. If a clause is falsified, conflict analysis derives a a new *learned clause*, identifies its *asserting literal*, and adds the clause to the solver's database. The solver then backjumps to the highest decision level where the learned clause becomes unit and enqueues the asserting literal. A conflict at decision level 0 establishes that the formula is UNSAT, while a complete assignment with no conflicts proves SAT.

**Branching Heuristic and Implementation.**  Branching is the central operation in CDCL: at each step of the search, the solver selects a variable and its polarity to branch on. The quality of these choices is critical. Strong decisions can greatly reduce the number of propagations and conflicts, whereas weak ones may cause the search to grow exponentially.

Usually, the choice of the next decision variable, together with its phase, is determined by a *branching heuristic*. The most influential family of heuristics is the *Variable State Independent Decaying Sum* (VSIDS) (Moskewicz et al., 2001). In VSIDS, variables are assigned activity scores that are increased whenever they appear in learned clauses. These scores are then periodically decayed, ensuring that recently relevant variables are favored for branching. This dynamic prioritization enables the solver to focus on the most promising parts of the search space. A notable implementation is MiniSAT (Eén & Sörensson, 2003), a lightweight and extensible open-source CDCL solver. Despite its minimalistic codebase, MiniSAT has become a standard platform for both research and industrial applications due to its clarity and effectiveness.

## 4   ImitSAT

As reviewed in Section 3, branching plays a pivotal role in CDCL: the quality of branching decisions largely determines the efficiency of unit propagation, which is the dominant cost in practice. Traditional heuristics such as VSIDS are highly effective but ultimately hand-crafted, leaving open the possibility of more principled approaches that can exploit structural patterns in solver traces. In this paper, we introduce ImitSAT, a framework that leverages expert-guided sequence modeling to learn high-quality branching policies for CDCL solvers.

Our approach consists of two main components: (i) *KeyTrace construction*, which compresses solver runs by collapsing backtracks into a concise sequence of expert decisions (Section 4.1); and (ii) an *autoregressive learner*, which

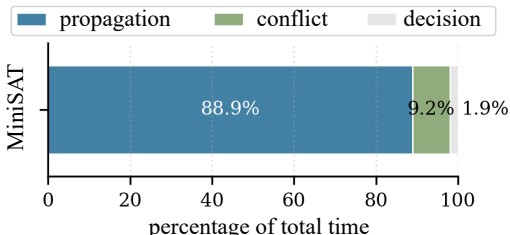

Figure 2: Propagation dominates CDCL time. MiniSAT spends about 88.9% in propagation, 9.2% in conflict analysis, and 1.9% in branching. Reducing propagation is therefore the main route to wall-clock gains.

imitates the KeyTrace to predict the next branching decision in sequence (Section 4.2). These components are seamlessly integrated into the CDCL framework. Together, they provide clean supervision signals and enable plug-and-play deployment.

## 4.1 EXPERT KEYTRACE REPLAY

We seek an expert that guides the solver toward effective branching decisions while avoiding wasted detours. Raw CDCL trails are often long and contain many decisions that are later undone by backtracking. An ideal expert should preserve only the decisions that survive and discard detours, thereby yielding clean training targets and eliminating steps that do not advance the search. In CDCL, unit propagation dominates runtime, typically accounting for 80%–90% of the total solving time (Zhang & Malik, 2002; Davis et al., 2008; Moskewicz et al., 2001), a trend we also confirm in Figure 2. This observation suggests that an expert which avoids wasted branches can substantially reduce propagation and thereby improve overall efficiency.

In order to extract expert supervision for branching, we first examine the execution trace of a CDCL solver. Each run on an instance $F$ produces a *trail*, namely a chronological sequence of decision, propagation, and backtracking events annotated with decision levels:

$$\mathcal{T}_t = ((\tau_1, \lambda_1, h_1), \ldots, (\tau_t, \lambda_t, h_t)) , \tag{5}$$

where $\tau_i \in \{\texttt{D}, \texttt{A}, \texttt{BT}\}$ representing a decision, a unit propagation assignment, or a backtrack event, respectively. $\lambda_i$ is a signed variable with $\lambda_i \in \{\pm 1, \ldots, \pm n\}$. We interpret $\lambda_i = +j$ as $x_j = \top$ and $\lambda_i = -j$ as $x_j = \bot$. The value $h_i$ is the decision level after the event. Raw trails $\mathcal{T}_t$ are often long and contain numerous events that are eventually undone by backtracking. Such redundant segments do not contribute to solving progress, yet they inflate computational cost and inject noise into the context, making it harder to isolate the key decisions that drive the search.

To remove these detours, we construct an expert *KeyTrace* through a systematic extraction procedure. Specifically, we scan $\mathcal{T}_t$ from left to right while maintaining a working sequence $\mathcal{K}$ that starts empty:

$$\mathcal{K} \leftarrow \emptyset. \tag{6}$$

For each event $(\tau_i, \lambda_i, h_i)$ we update

$$\mathcal{K} \leftarrow \begin{cases} \mathcal{K} \parallel (\texttt{D}, \lambda_i, h_i), & \text{if } \tau_i = \texttt{D}, \\ \mathcal{K} \parallel (\texttt{A}, \lambda_i, h_i), & \text{if } \tau_i = \texttt{A}, \\ \text{trim}_{\leq h_i}(\mathcal{K}) \parallel (\texttt{D}, \lambda_i, h_i), & \text{if } \tau_i = \texttt{BT}, \end{cases} \tag{7}$$

where $\parallel$ denotes concatenation. The operator $\text{trim}_{\leq h}$ removes all suffix events whose level is above $h$. Restarts are handled by trimming to level 0, i.e., $\mathcal{K} \leftarrow \text{trim}_{\leq 0}(\mathcal{K})$. After the scan, the resulting sequence

$$\mathcal{K}_t = \mathcal{K} \tag{8}$$

is taken as the expert KeyTrace for trail $\mathcal{T}_t$.

It is worth noting that the above procedure preserves only the decision and propagation events that survive backtracking, effectively collapsing backtracks into prefix truncations while treating restarts as full resets. This results in a considerably shorter and more stable trail representation, capturing the minimal necessary context for advancing the search.

Empirical evidence demonstrates that replaying the decision sequence encoded in $\mathcal{K}_t$ on the same problem instance renders the solver's run nearly conflict-free, drastically reducing the number of propagation events to approximately 4% of those recorded in the raw MiniSAT trail ( see Figure 3). Since propagation constitutes the dominant runtime component of CDCL solvers, this reduction translates directly into significant improvements in solver efficiency.

## 4.2 AUTOREGRESSIVE IMITATION FOR CDCL BRANCHING

Given this compact expert KeyTrace, our goal is to train a learner that can imitate expert-quality branching decisions. In this setting, the expert demonstrations are provided by KeyTraces: each surviving decision, together with its implied propagations, forms a supervised training target. Unlike reinforcement-style exploration, this approach offers dense, step-level supervision and eliminates

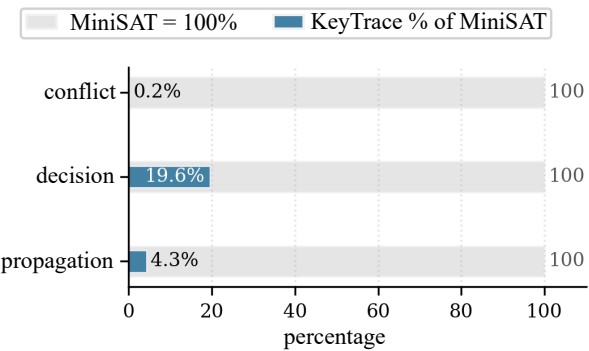

Figure 3: Compact KeyTrace replay. The expert replayed only a small share of MiniSAT events: 0.2% conflicts, 19.6% decisions, and 4.3% propagations.

detours, allowing the learner to acquire high-quality decision policies with faster convergence and greater stability.

Formally, since a CDCL solver queries one branch at a time, the learning task is to map the formula $F$ together with a KeyTrace prefix $\mathcal{K}_t$ to the next signed variable, while operating under a small computational budget. This task is inherently sequential, as each decision depends on the prefix of earlier assignments. An autoregressive (AR) model is therefore a natural choice, as it conditions on the serialized prefix and predicts the next element in the sequence. To realize this, we first serialize the solver state into a compact sequence tailored for autoregression, then specify the AR next-decision learner and its behavior-cloning objective, and finally describe how the learner is integrated into CDCL at inference time.

**Serialization.** To make the AR learner effective, both the CNF $F$ and the KeyTrace prefix $\mathcal{K}_t$ are serialized into one deterministic sequence. The CNF is written in DIMACS integer format $F_{\text{DIMACS}}$, followed by separator tokens [SEP]. The KeyTrace is then presented as blocks, where each decision literal is followed by the unit assignments produced by propagation. Level fields and assignment tags are omitted for compactness. The sequence ends with a decision probe marker [D] that signals a request for the next branch. Formally, with $\mathcal{K}_t = \big((\tau_1, \lambda_1, h_1), \ldots, (\tau_t, \lambda_t, h_t)\big)$,

$$\text{enc}(\mathcal{K}_t) = \big\|_{i=1}^{J} (\text{D}, d_i, a_{i,1}, \ldots, a_{i,k_i}), \qquad d_i, a_{i,k} \in \{\pm 1, \ldots, \pm n\}, \tag{9}$$

and the full serialized input is

$$z(F, \mathcal{K}_t) = [\text{CNF}] \, \| \, F_{\text{DIMACS}} \, \| \, [\text{SEP}] \, \| \, \text{enc}(\mathcal{K}_t) \, \| \, [\text{D}]. \tag{10}$$

This compact representation aligns exactly with next-decision prediction for an AR learner.

**Next-decision AR learner.** The learner implements the policy

$$p_\theta(\lambda_{\text{next}} \mid F, \mathcal{K}_t), \qquad \lambda_{\text{next}} \in \{\pm 1, \ldots, \pm n\}, \tag{11}$$

using an AR model defined over $z(F, \mathcal{K}_t)$. For a generic sequence $x_1, \ldots, x_S$, the AR factorization is

$$p_\theta(x_1, \ldots, x_S) = \prod_{s=1}^{S} p_\theta(x_s \mid x_{<s}), \tag{12}$$

where $s$ is the index immediately following the decision probe marker [D]. Under this serialization, predicting the next CDCL decision is exactly next-symbol prediction at position $s$:

$$p_\theta(\lambda_{\text{next}} \mid F, \mathcal{K}_t) = p_\theta(x_s = \lambda_{\text{next}} \mid x_{<s} = z(F, \mathcal{K}_t)). \tag{13}$$

Training follows the standard *behavior cloning* paradigm in imitation learning. Let $\mathsf{Q}$ be the set of triples $(F, \mathcal{K}_t, d_{t+1})$. Training minimizes the average negative log-likelihood of the expert decisions:

$$\mathcal{L}(\theta) = \frac{1}{|\mathsf{Q}|} \sum_{(F, \mathcal{K}_t, d_{t+1}) \in \mathsf{Q}} \Big[ -\log p_\theta(d_{t+1} \mid z(F, \mathcal{K}_t)) \Big], \tag{14}$$

which is equivalent to cross-entropy at the decision probe positions. In other words, the learner directly imitates the expert policy encoded in the KeyTraces.

**Online integration into CDCL.**  At each decision point, the learner is queried under a small budget. The current trail is collapsed into the KeyTrace, the instance and prefix are serialized, and the model predicts one signed variable. If the prediction is legal (*i.e.,* within the variable range and currently unassigned), the solver accepts it and decrements the query budget. Otherwise, the solver immediately falls back to VSIDS heuristic. This design preserves completeness and keeps overhead small. A front-loaded query schedule is particularly effective, as early decisions strongly shape most of the search as shown in Appendix E. A complete Algorithm 1 description is provided in Appendix A.

## 5 EXPERIMENTS

In this section, we evaluate ImitSAT as a learned branching policy for CDCL solvers. We first describe the experimental setup, including baselines, solver implementation, datasets, model, and evaluation metrics. We then compare ImitSAT with SATformer and Graph-Q-SAT on random 3-SAT test sets and on a range of structured SAT families. Wall clock behaviour is studied both in Section 5.4 and through a direct comparison with pure MiniSAT in Appendix O, and we also examine generalization to industrial benchmarks from SATCOMP in Appendix M. Finally, we summarize several additional analyses whose full results are deferred to the appendices, including a query budget ablation, a GNN augmented variant, a Top-K masking versus fallback study, and integrations with more advanced solvers such as CaDiCaL (Biere et al., 2024a) and Kissat (Biere et al., 2024b), detailed in Appendices N, P, Q, and R.

### 5.1 IMPLEMENTATION DETAILS

**Baselines.**  We compare against SATformer (Shi et al., 2023) and Graph-Q-SAT (Kurin et al., 2020), which are trained using their public implementations and specified training data. We also retrain them on our dataset, with equivalent training volume, denoted as SATformer* and GQSAT*.

**CDCL Solver.**  A Python reimplementation of MiniSAT 2.2 (Eén & Sörensson, 2003) is used and validated against the official C++ version on full trails $\mathcal{T}_t$. This version streamlines the integration of learning methods, ensuring that all approaches are evaluated in a consistent environment.

**Datasets.**  We generate the random 3–SAT training dataset with a planted assignment (Mezard & Montanari, 2009; Achlioptas et al., 2000; 2005). The clause–variable ratio is in $[4.1, 4.4]$. Variable ranges include 5 to 15, 16 to 30, 31 to 60, and 61 to 100, with additional fixed sizes of 50 and 100. For each formula, a MiniSAT run yields the level-annotated trail as Equation 5. Then, the expert KeyTrace is extracted using Equation 7. Every decision position provides one supervision pair, a KeyTrace prefix $\mathcal{K}_t$, and its next decision $d_{t+1}$. For evaluation, we first use the test set, which is drawn from the same generators and variable ranges as those used for training. Generalization is then assessed on different SAT families from SATLIB (Hoos & Stützle, 2000). Further dataset details appear in Appendix B.

**AR learner and training.**  Perceiver AR (Hawthorne et al., 2022) serves as our architecture for predicting the next decision. Specifically, the model uses an output latent array that cross-attends to the input. Since only one output is needed for the next branching, we set the latent length to 1. This results in each query having $O(N)$ complexity in terms of input length, avoiding the $O(N^2)$ cost associated with a standard Transformer decoder. Furthermore, our model configuration follows the recommendations in (Hawthorne et al., 2022). We use 16 attention heads for cross-attention and self-attention, 12 Transformer blocks, an MLP expansion of $4\times$, squared-ReLU activations, and cross-attention dropout $0.1$.

**Metrics.**  Propagation dominates CDCL time (Zhang & Malik, 2002; Davis et al., 2008; Moskewicz et al., 2001), as seen in Figure 2 and Table 6. We therefore use the number of propagation as the primary signal. Since instances vary in size and difficulty, we normalize by MiniSAT on a per-instance basis and aggregate with the median to reduce the influence of outliers.

Table 1: MRPP $\tilde{r}$ ($\downarrow$) and one percent win ratio $W_{1\%}$ ($\uparrow$) on 3-SAT test sets. GQSAT denotes Graph-Q-SAT.

| Metric | Method | 5–15 | 16–30 | 31–60 | 61–100 | 50 | 100 |
|---|---|---|---|---|---|---|---|
| MRPP $\tilde{r}$ ($\downarrow$) | KeyTrace | 0.57 | 0.39 | 0.21 | 0.10 | 0.17 | 0.03 |
| | GQSAT-3calls | 1.00 | 0.94 | 0.89 | 1.15 | 0.71 | 0.85 |
| | GQSAT-5calls | 1.00 | 0.90 | 0.82 | 0.94 | 0.70 | 0.80 |
| | SATformer | 1.00 | 0.89 | 0.84 | **0.78** | 0.88 | 0.81 |
| | GQSAT*-3calls | 1.38 | 1.36 | 1.46 | 1.59 | 1.37 | 1.41 |
| | GQSAT*-5calls | 1.43 | 1.44 | 1.53 | 1.51 | 1.23 | 1.40 |
| | SATformer* | 1.00 | 0.88 | 0.80 | 0.83 | 0.86 | 0.82 |
| | Ours-3calls | 0.75 | 0.83 | **0.75** | **0.78** | 0.74 | **0.76** |
| | Ours-5calls | **0.73** | **0.77** | **0.75** | 0.80 | **0.66** | 0.83 |
| $W_{1\%}$ ($\uparrow$) | KeyTrace | 0.71 | 0.87 | 0.96 | 0.98 | 0.97 | 0.99 |
| | GQSAT-3calls | 0.45 | 0.53 | 0.54 | 0.48 | 0.57 | 0.55 |
| | GQSAT-5calls | 0.46 | 0.54 | 0.56 | 0.53 | 0.59 | 0.58 |
| | SATformer | 0.48 | 0.55 | 0.58 | 0.57 | 0.60 | 0.57 |
| | GQSAT*-3calls | 0.25 | 0.33 | 0.34 | 0.38 | 0.38 | 0.36 |
| | GQSAT*-5calls | 0.23 | 0.30 | 0.35 | 0.34 | 0.37 | 0.36 |
| | SATformer* | 0.48 | 0.56 | 0.58 | 0.53 | 0.57 | 0.56 |
| | Ours-3calls | **0.68** | **0.65** | **0.65** | **0.64** | **0.69** | **0.60** |
| | Ours-5calls | 0.67 | 0.64 | 0.61 | 0.59 | 0.64 | 0.56 |

Let $p_i$ be the propagation count of the evaluated method on instance $F_i$, and let $p'_i$ be the corresponding count for MiniSAT. We report the *Median Relative Propagation Percentage* (MRPP)

$$\tilde{r} = \mathrm{med}_{i=1,\ldots,N}\left(\frac{p_i}{p'_i}\right), \tag{15}$$

values $\tilde{r} < 1$ indicate fewer propagation than MiniSAT and hence an improvement.

To capture instance-wise gains, we also report a one-percent win rate. With margin $\delta = 0.01$

$$W_\delta = \frac{1}{N} \sum_{i:\, p'_i > 0} \mathbf{1}\left[p_i \leq (1 - \delta)\, p'_i\right]. \tag{16}$$

This is the fraction of problems on which the method reduces the propagation number by at least 1% relative to MiniSAT; therefore, a larger $W_{1\%}$ is better.

## 5.2 COMPARISON ON TEST SETS

We compare ImitSAT with Graph-Q-SAT and SATformer on held-out random 3-SAT. To align wall-clock budgets, ImitSAT is queried at 3 or 5 times per instance. Graph-Q-SAT uses the same computational budgets. SATformer updates the VSIDS variable scores once at initialization. We report MRPP $\tilde{r}$ and the one percent win rate $W_{1\%}$. We also include the GQSAT* and SATformer* variants, which are trained on our random 3-SAT dataset to have a fair comparison. On the test set, GQSAT* performs much worse than GQSAT, while SATformer* remains close to SATformer.

Table 1 (top) shows that ImitSAT achieves the lowest MRPP $\tilde{r}$ in nearly all ranges. With 3 or 5 calls, it is best on 5–15, 16–30, 31–60, 50, 100, and ties SATformer on 61–100. These results indicate that imitation of the expert, as demonstrated by KeyTrace, consistently reduces propagation under small query budgets. Building on these MRPP results, Graph-Q-SAT increases propagation number on the 61-100 dataset, indicating weaker guidance under the same budget. Finally, in Table 1 (bottom), we report $W_{1\%}$ to show instance-wise gains. ImitSAT with 3 calls achieves the highest win rate in all ranges.

Table 2: MRPP $\tilde{r}$ ($\downarrow$) and one percent win ratio $W_{1\%}$ ($\uparrow$) on structured SAT families.

| Metric | Method | JNH | AIM | PARITY | PHOLE | PRET |
|--------|--------|-----|-----|--------|-------|------|
| MRPP $\tilde{r}$ ($\downarrow$) | KeyTrace | 0.18 | 0.55 | 0.11 | 0.97 | 0.56 |
| | GQSAT-3calls | 1.29 | 1.20 | 0.82 | 1.03 | 0.88 |
| | GQSAT-5calls | 1.11 | 1.18 | 0.56 | 0.82 | 0.92 |
| | SATformer | 1.36 | 1.01 | 0.73 | 1.00 | 1.00 |
| | GQSAT*-3calls | 1.39 | 1.15 | 0.66 | 1.05 | 1.00 |
| | GQSAT*-5calls | 1.84 | 0.82 | 0.51 | **0.77** | 0.54 |
| | SATformer* | 1.75 | 0.95 | 0.73 | 1.00 | 1.00 |
| | Ours-3calls | 1.00 | 0.88 | 0.30 | 1.00 | **0.42** |
| | Ours-5calls | **0.85** | **0.81** | **0.30** | 0.82 | **0.42** |
| $W_{1\%}$ ($\uparrow$) | KeyTrace | 1.00 | 0.75 | 1.00 | 0.50 | 1.00 |
| | GQSAT-3calls | 0.38 | 0.31 | **0.80** | 0.50 | 0.50 |
| | GQSAT-5calls | 0.44 | 0.38 | **0.80** | 0.75 | 0.50 |
| | SATformer | 0.25 | 0.44 | 0.60 | 0.00 | 0.00 |
| | GQSAT*-3calls | 0.38 | 0.31 | 0.60 | 0.50 | 0.25 |
| | GQSAT*-5calls | 0.31 | 0.56 | **0.80** | 0.75 | 0.75 |
| | SATformer* | 0.13 | 0.50 | 0.60 | 0.00 | 0.00 |
| | Ours-3calls | 0.44 | **0.63** | **0.80** | 0.50 | **1.00** |
| | Ours-5calls | **0.50** | **0.63** | **0.80** | 0.75 | **1.00** |

## 5.3 GENERALIZATION ON SPECIAL SAT FAMILIES

In this section, we assess the ability to transfer to structured families that differ from the training generator, covering SAT and UNSAT, 3-SAT, and non-$k$-SAT, without any retraining or tuning. Query budgets match the test sets, with 3 or 5 calls per instance for ImitSAT and Graph-Q-SAT, whereas SATformer adjusts the VSIDS variable scores only once at initialization.

Across all families, ImitSAT attains the lowest MRPP $\tilde{r}$ or ties for best as shown in Table 2 (top). The gains are large on PARITY and PRET, and with 5 calls ImitSAT is best on JNH and AIM, and matches the best on PHOLE. Graph-Q-SAT does not reduce propagation number on JNH and AIM, and in fact increases the propagation number. SATformer only shows a gain on the PARITY dataset; all other datasets exhibit no gain or worse performance than MiniSAT. Table 2 (bottom) reports the $W_{1\%}$. ImitSAT achieves the highest or tied win rate on all families, including a perfect score on PRET and matching the best PARITY and PHOLE. These outcomes indicate that imitation of the expert KeyTrace transfers from random 3-SAT to structured regimes without finetuning. We report additional experiments on industrial benchmarks from SATCOMP (Iser & Jabs, 2024) in Appendix M, using instances with at most 100 variables and a DIMACS encoding that fits within our context budget. On this filtered subset, ImitSAT and SATformer solve the harder instances substantially faster in wall-clock time than Graph-Q-SAT under the same query budget as shwon in Figure 6.

## 5.4 WALL-CLOCK TIME

To measure practical impact, end-to-end solve time is recorded for each instance. The timer starts when the CDCL solve loop begins and stops when the instance is solved; CNF parsing and simplification are excluded from the timing. All model inference costs are included. ImitSAT and Graph-Q-SAT receive 3 calls per instance to match compute budgets. SATformer adjusts VSIDS variable scores once at initialization.

Across random 3-SAT test sets and structured families, ImitSAT traces the lowest curves under these budgets, which means more instances are solved in less time, as shown in Figure 4. Learning model-based branching introduces query overhead, so wall-clock gains appear only once propagation savings exceed this cost. We also compare ImitSAT directly with pure MiniSAT on 16–30, 31–60, and 61–100, as summarized in Table 12 and detailed in Appendix O. On the easier 16–30

and 31–60 ranges, MiniSAT remains faster while ImitSAT is the strongest learned method and stays close in runtime; on the harder 61–100 range, ImitSAT achieves the lowest wall-clock time overall, showing that its propagation savings translate into a net speedup once instances are sufficiently large and challenging.

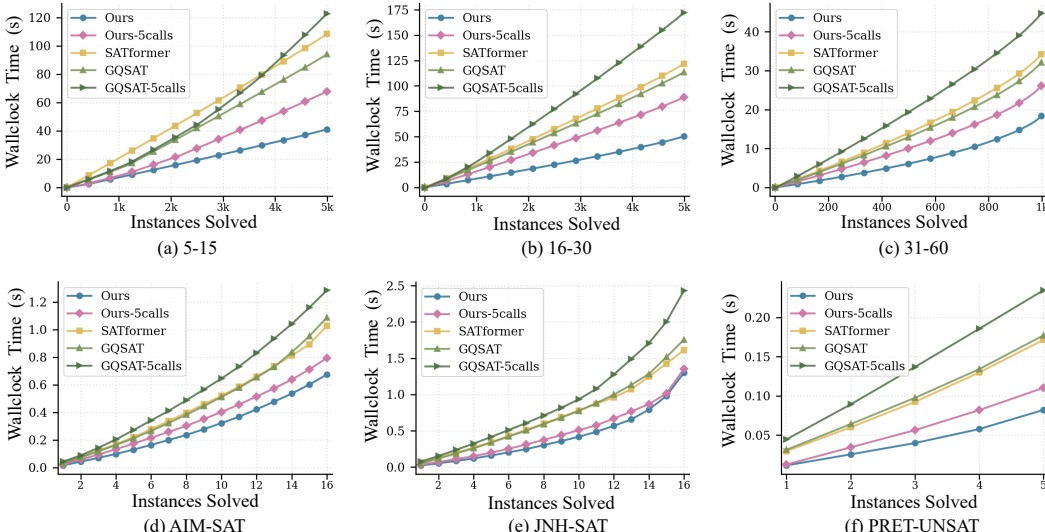

Figure 4: Wall-clock time versus instances solved on random test sets and structured families. Curves show end-to-end solver time per instance with all model calls included and preprocessing excluded. ImitSAT and Graph-Q-SAT use 3 or 5 calls per instance. SATformer performs a single VSIDS initialization. The lower curve is better.

## 5.5 IMPROVED TRAINING TECHNIQUES

We observe that ImitaSAT is prone to overfitting. To address this issue, we introduce improved training techniques. First, we apply variable permutation augmentation, where variable IDs are randomly permuted when forming training examples. As shown in Figure 5, both training and validation losses decay steadily with augmentation, whereas without augmentation the validation loss peaks early and then rises despite continued decrease in training loss. Second, we employ a staged curriculum learning strategy that gradually expands the variable range from small to large (Nagatsuka et al., 2021; Pouransari et al., 2024). This accelerates convergence on simple instances while ensuring comprehensive coverage of larger problem sizes. More details for variable permutation augmentation and curriculum learning are provided in Appendix F and G.

## 6 CONCLUSION

We presented ImitSAT, a CDCL branching policy based on imitation learning. By collapsing solver runs into compact sequences of surviving decisions, we obtain expert traces that capture high-quality branching behavior. These traces allow us to formulate branching as an autoregressive prediction problem, yielding dense, conflict-free supervision that enables stable and efficient training. Extensive experiments show that ImitSAT reduces propagations, achieves favorable runtime, and generalizes well beyond 3-SAT, outperforming prior learning-based methods. Future work could extend this approach to richer expert demonstrations, hybrid imitation–reinforcement learning schemes, or broader domains of combinatorial reasoning.

## ACKNOWLEDGMENTS

X. Xu acknowledges support from the National Natural Science Foundation of China (62302385).

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

# A    ALGORITHM

The integration of the CDCL and ImitSAT is shown in Algorithm 1.

---

**Algorithm 1** Online CDCL branch with ImitSAT.

---

**Require:** CNF $F$, learner $p_\theta$, total query budget $B$
**Ensure:** terminal outcome (SAT/UNSAT) or next branching decision $\lambda_{\text{next}}$

1: $b \leftarrow B$
2: $\mathcal{T}_t \leftarrow \text{CURRENTTRAIL}()$           ▷ level-annotated trail as in Equation 5
3: **if** $\text{CONFLICTATLEVELZERO}(\mathcal{T}_t)$ **then**
4:     **return** UNSAT
5: **end if**
6: **if** $\text{ALLASSIGNED}(\mathcal{T}_t)$ **then**
7:     **return** SAT
8: **end if**
9: $\mathcal{K}_t \leftarrow \text{EXTRACTKEYTRACE}(\mathcal{T}_t)$         ▷ collapse backtracks, Equation 7
10: **if** $b > 0$ **then**
11:     $b \leftarrow b - 1$           ▷ consume one model query
12:     $\lambda_{\text{next}} \leftarrow \arg\max_\lambda \; p_\theta(\lambda \mid z(F, \mathcal{K}_t))$
13:     **if** $\text{LEGAL}(\lambda_{\text{next}}; F, \mathcal{T}_t)$ **then**     ▷ unassigned and within the variable range
14:         **return** $\lambda_{\text{next}}$
15:     **end if**
16: **end if**
17: **return** $\text{VSIDS}(F, \mathcal{T}_t)$           ▷ fallback decision

---

# B    DATASETS

This section details the training corpus, held-out test sets, and the structured SAT families used in evaluation. For the synthetic data, both train and test, all instances are random 3-SAT with a planted assignment and a clause–variable ratio in $[4.1, 4.4]$ (Mezard & Montanari, 2009; Achlioptas et al., 2000; 2005). We categorize instances by the number of variables $n_v$ to control difficulty and balance the number of decision probes contributed by each category. Since larger $n_v$ produces longer trails and thus more supervision per instance, we allocate fewer instances at larger $n_v$. This ensures each bucket contributes a similar amount of training signal. We define buckets as ranges of variable counts (e.g., "5–15"), with each bucket grouping instances by the number of variables $n_v$ they contain.

**Training dataset.**    Counts denote the number of CNF instances per bucket.

Table 3: Training buckets for random 3-SAT with a planted assignment.

| Bucket (variables $n_v$) | # Instances |
|:---:|:---:|
| 5–15 | 2,000,000 |
| 16–30 | 1,000,000 |
| 31–60 | 500,000 |
| 61–100 | 100,000 |
| 50 | 1,000,000 |
| 100 | 100,000 |

**Held-out test buckets.**    To keep evaluation time reasonable, we use fewer instances at larger $n_v$. For $n_v{=}100$, we increase the sample size to reduce variance.

Table 4: Held-out test buckets for random 3–SAT.

| Bucket (variables $n_v$) | # Instances |
|---|---|
| 5–15 | 5,000 |
| 16–30 | 5,000 |
| 31–60 | 1,000 |
| 61–100 | 100 |
| 50 | 100 |
| 100 | 500 |

**Structured families from SATLIB.** To assess cross-distribution generalization, we evaluate on classic structured benchmarks from SATLIB (Hoos & Stützle, 2000). The suite spans both SAT and UNSAT, and includes 3-SAT and non–$k$–SAT regimes. Specifically, we use AIM (Asahiro et al., 1996), which is equivalent to 3-SAT, focusing on the $n_v = 100$ SAT portion. We also consider the SAT subset of JNH (Selman et al., 1996) with $n_v = 100$ and PARITY (Warners & Van Maaren, 1998; Hoos & Stützle, 2000) in the compressed series, both as structured SAT families distinct from our random generator. Additionally, PRET (Johnson & Trick, 1996; Hoos & Stützle, 2000) is a two-coloring UNSAT problem with $n_v = 60$, and the pigeonhole principle (PHOLE) (Haken, 1985) is used as a non–$k$–SAT and UNSAT stress test. All subsets described above are considered with $n_v \leq 100$. Table 5 lists the families.

Table 5: The structured families referenced in our generalization experiments are all sourced from SATLIB.

| Family | $k$–SAT ? | SAT/UNSAT |
|---|---|---|
| AIM | 3–SAT | SAT |
| JNH | non–$k$–SAT | SAT |
| PARITY | non–$k$–SAT | SAT |
| PHOLE | non–$k$–SAT | UNSAT |
| PRET | non–$k$–SAT | UNSAT |

## C  CDCL RUNTIME BREAKDOWN

To quantify the main cost drivers in CDCL, we instrument the Python MiniSAT on the 61–100 test range and time each major component. Table 6 and Figure 2 show that propagation accounts for about 89% of the mean runtime, conflict analysis for about 9%, and decision selection for under 2%. Reducing needless propagation is therefore the most direct path to wall-clock gains.

Table 6: The CDCL runtime breakdown on the 61–100 test set. Propagation dominates the runtime, motivating metrics and methods that reduce propagation.

| Event | mean ms | median ms | share of mean % |
|---|---|---|---|
| Propagation | 65.624 | 38.378 | 88.88 |
| Conflict | 6.809 | 3.999 | 9.22 |
| Decision | 1.400 | 0.975 | 1.90 |

## D  KEYTRACE REPLAY

We confirm KeyTrace is a suitable expert because replaying it yields a near conflict-free run with far fewer propagations. To demonstrate its effectiveness, for each instance, we replay the extracted KeyTrace by applying its decisions in order and running unit propagation after each step. As shown in Table 7 and Figure 3, the event counts during the KeyTrace replay can be compared to those of the

original MiniSAT run. Notably, conflicts are essentially eliminated, decisions drop by about 80%, and propagations fall to roughly 4% of the MiniSAT total. Together, these results support using KeyTrace as the expert for behavior cloning: it isolates the surviving branch sequence and removes the detours that drive most propagation.

Table 7: Effect of KeyTrace replay on the 61–100 range. Means over instances and the share relative to MiniSAT. Replay is nearly conflict-free and uses only a small fraction of propagation.

| Event | MiniSAT mean | KeyTrace mean | KeyTrace as % of MiniSAT |
|---|---|---|---|
| Conflict | 71.96 | 0.11 | 0.15% |
| Decision | 103.94 | 20.38 | 19.61% |
| Propagation | 1474.64 | 62.74 | 4.25% |

# E  EARLY DECISIONS SHAPE CDCL

In this section, we study how the timing of model queries affects the overall CDCL search.

We first compare two methods that use a single model call and have identical compute budgets on the 5–15 test set: (1) querying ImitSAT at the very first branching decision (call at first decision), and (2) making three branching decisions using the VSIDS heuristic before making a single model query (call after 3 VSIDS). We report MRPP $\tilde{r}$ and $W_{1\%}$ in Table 8, showing that delaying the single query (2) does not reduce propagations and substantially decreases the win rate, whereas front-loading the query (1) improves both metrics. This supports allocating the model budget to early decisions, as they exert a strong influence on the subsequent search.

To further investigate this effect on larger datasets, we evaluate three query strategies on the 100 test set and AIM: (a) three model calls after completing three VSIDS decisions (3 calls after 3 VSIDS), (b) three model calls, each separated by three VSIDS decisions (3 calls, 1 per 3 VSIDS), and (c) three model calls issued immediately at the start of the search (3 calls at first decision). All subsequent decisions in each case use the native VSIDS heuristic. In Table 9, method (c) achieves the best MRPP $\tilde{r}$ and $W_{1\%}$, indicating that a simple greedy policy that spends the budget as early as possible is already a strong baseline.

Table 8: Early guidance is more effective. One model calls on the 5–15 test set. Front-loading the call at the first decision yields larger gains than delaying it.

| Method | $\tilde{r}$ ($\downarrow$) | $W_{1\%}$ ($\uparrow$) |
|---|---|---|
| call after 3 VSIDS | 1.00 | 0.27 |
| call at first decision | **0.91** | **0.55** |

Table 9: Comparison of query heuristics. Using the model greedily at the beginning of the search is much more effective than distributing calls across later VSIDS decisions.

| Metric | Method | 100 | AIM |
|---|---|---|---|
| | 3 calls after 3 VSIDS | 0.87 | 1.02 |
| MRPP $\tilde{r}$ ($\downarrow$) | 3 calls, 1 per 3 VSIDS | 0.95 | 1.09 |
| | 3 calls at first decision | **0.76** | **0.88** |
| | 3 calls after 3 VSIDS | 0.57 | 0.13 |
| $W_{1\%}$ ($\uparrow$) | 3 calls, 1 per 3 VSIDS | 0.52 | 0.19 |
| | 3 calls at first decision | **0.60** | **0.63** |

## F    VARIABLE PERMUTATION AUGMENTATION

Permutation augmentation is designed to mitigate overfitting and enhance the robustness of the learner. We train models on 5–15 variables for 20 epochs, or about 300k steps. Figure 5 plots the training and validation losses. With permutation augmentation, the two curves track each other and decay steadily. Without augmentation, the training loss continues to decrease, while the validation loss peaks early and then rises, a classic sign of overfitting to variable identities. The aggregate metrics confirm this, as in Table 10 that removing permutation leads to a higher MRPP $\tilde{r}$ and a low win rate, whereas the augmented model achieves a strong MRPP $\tilde{r}$ and a much higher win rate $W_{1\%}$. The slightly higher training loss under augmentation is expected, as the task is more challenging; however, the gain is evident in generalization.

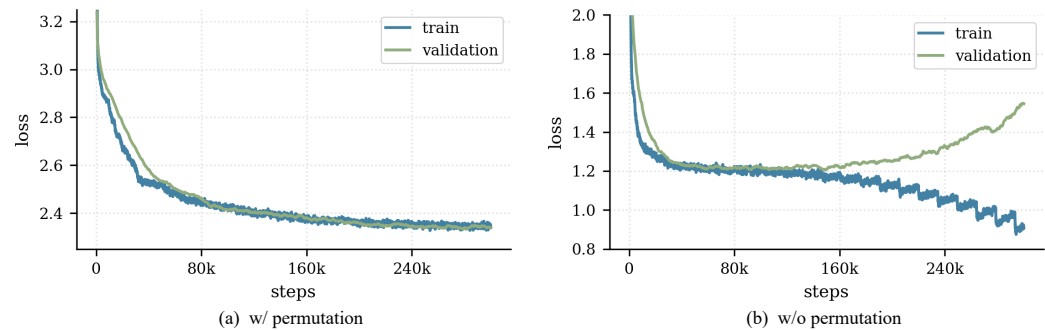

(a)  w/ permutation                            (b)  w/o permutation

Figure 5: Training and validation loss curves with and without variable-permutation augmentation on the 5–15 range for 20 epochs. Permutation keeps training and validation closely aligned and prevents overfitting, while removing it lowers training loss but drives validation loss up.

Table 10: Effect of variable-permutation augmentation. Models trained on 5–15 for 20 epochs. Augmentation prevents overfitting and improves test set metrics.

| Method | $\tilde{r}$ ($\downarrow$) | $W_{1\%}$ ($\uparrow$) |
|---|---|---|
| w/o permutation | 1.00 | 0.28 |
| w/ permutation | 0.75 | 0.64 |

## G    STAGED CURRICULUM ACROSS VARIABLE RANGES

The staged curriculum is designed to accelerate learning and maintain competence across a wide range of variables (Nagatsuka et al., 2021; Pouransari et al., 2024). The curriculum trains on 5–15 for 20 epochs, then continues on 16–30 for 4 epochs, about 130k steps in the second stage. Two baselines are used on 16–30 without any curriculum. The first matches the second-stage budget by only about 130k steps. The second uses a total of 430k steps that match the curriculum training steps.

Table 11 summarizes the results. At the matched 130k steps budget on 16–30, the staged model achieves a much lower $\tilde{r}$ and a higher $W_{1\%}$ than training the model "w/o stage" from scratch; this shows clear sample efficiency. When step counts are fully matched as "w/o stage*", training from scratch on 16–30 narrows the gap and slightly edges out the staged model in that range. This suggests that the primary benefit of the curriculum is faster and more stable convergence, rather than better performance. Crucially, when the curriculum progresses through all stages, it preserves strong performance on 5–15, whereas models trained only on 16–30 fail to generalize to smaller instances.

Table 11: Stage-training ablation. Ours uses a curriculum that trains on 5–15 for 300k steps, then continues on 16–30 for 130k steps. The *w/o stage* trains only on 16–30 with the same second-stage budget, about 130k steps. And *w/o stage\** trains only on 16–30 with a total of about 430k steps that matches the overall curriculum steps. Curriculum improves sample efficiency and yields better coverage of the 5–15 range.

| Method | 5-15 $\tilde{r}$ ($\downarrow$) | 5-15 $W_{1\%}$ ($\uparrow$) | 16-30 $\tilde{r}$ ($\downarrow$) | 16-15 $W_{1\%}$ ($\uparrow$) |
|---|---|---|---|---|
| w/o stage | 1.00 | 0.05 | 1.00 | 0.49 |
| w/o stage* | 1.00 | 0.43 | 0.80 | 0.66 |
| w/ stage | 0.75 | 0.67 | 0.83 | 0.63 |

## H KEYTRACE EXAMPLE

This section walks through Figure 1 on an example instance, showing a short CDCL run and how it collapses into a KeyTrace. Consider the CNF $F$ over $x_1, x_2, x_3, x_4$,

$$F = (x_1 \lor x_2 \lor \neg x_3) \land (\neg x_4 \lor \neg x_2 \lor \neg x_3) \land (x_1 \lor x_3 \lor \neg x_4 \lor x_2)$$
$$\land (\neg x_3 \lor \neg x_1 \lor \neg x_4) \land (x_3 \lor \neg x_4 \lor \neg x_2) \land (\neg x_2 \lor x_4 \lor x_3). \tag{17}$$

**One CDCL trail.** A CDCL run interleaves decisions (D), unit propagations (A), and backtracks (BT), each annotated with the decision level. One plausible trail is

$$\mathcal{T} = \big((\text{D}, +4, 1), (\text{D}, +3, 2), (\text{BT}, -3, 1),$$
$$(\text{BT}, -4, 0), (\text{D}, +1, 1), (\text{D}, +2, 2), (\text{A}, -3, 2)\big) \tag{18}$$

It can be read in three stages, matching the left panel of Figure 1. *Stage 1:* branch $x_4 = \top$, then $x_3 = \top$; a conflict is reached and the solver backtracks, forcing $x_3 = \bot$ at a lower level. *Stage 2:* a second conflict occurs and the run backtracks to level 0, flipping the earlier choice to $x_4 = \bot$. *Stage 3:* branch $x_1 = \top$ and $x_2 = \top$; unit propagation assigns $x_3 = \bot$, and all clauses are satisfied, so $F$ is declared SAT.

**Collapsing to a KeyTrace.** Applying the extraction rule in Equation 7 trims away backtracked suffixes and keeps only the surviving root–to–current decisions. For the trail above, the resulting expert KeyTrace is

$$\mathcal{K} = \big((\text{BT}, -4, 0), (\text{D}, +1, 1), (\text{D}, +2, 2), (\text{A}, -3, 2)\big), \tag{19}$$

i.e., the final branch $x_4 = \bot \to x_1 = \top \to x_2 = \top$ and then propagate $x_3 = \bot$ shown in the right panel of Figure 1. Replaying $\mathcal{K}$ on the same instance with unit propagation between steps is nearly conflict-free and avoids the detours taken in the original run, which is why KeyTrace provides clean targets for imitation.

## I LARGE LANGUAGE MODELS (LLMS) USAGE STATEMENT

We used LLMs as general-purpose assistants to scaffold small analysis or plotting scripts, suggest debugging tips, and polish wording. All technical ideas, algorithms, model designs, experiments, and reported results are the sole responsibility of the authors. LLM outputs were reviewed and edited for correctness and clarity. No proprietary data were provided to LLMs, and LLMs are not authors.

## J ETHICS STATEMENT

This work uses procedurally generated SAT instances and public SATLIB benchmarks; no human subjects or personally identifiable information are involved. The research poses minimal foreseeable risks to society. We will release code and models under a permissive license to encourage transparent and responsible use.

## K    REPRODUCIBILITY STATEMENT

We will release the following: (i) pretrained models; (ii) our Python reimplementation of Mini-iSAT 2.2, as well as the integration code for ImitSAT; (iii) training code, including all hyperparameters; (iv) generators for random planted 3-SAT data, along with the exact train/test datasets; and (v) environment specifications. The dataset definitions and splits are detailed in Appendix B.

## L    LIMITATIONS

The study was limited by computational resources, as all models were trained on four V100 GPUs, which restricted both model size and training duration. To address these constraints, staged curriculum learning was used to accelerate training. With greater access to GPUs, it would be possible to train models for longer periods on larger datasets and scale model size, potentially leading to improvements in imitation quality and wall-clock performance.

## M    COMPARISON ON SATCOMP

To gauge performance on industrial-style problems, we also benchmark on a subset of SATCOMP instances (Iser & Jabs, 2024) that fits our current text-based interface. We focus on formulas with up to 100 variables whose DIMACS representation fits within the model context, so that the entire CNF can be passed to the policy without truncation. This yields a set of real competition benchmarks with nontrivial structure that are compatible with our current model scale. Within this regime, we run MiniSAT, SATformer, GQSAT, and ImitSAT under the same query budgets as in the main experiments, capping MiniSAT at 10 seconds per instance to keep the evaluation tractable. Figure 6 reports mean wall-clock time over batches of 30 solved instances.

For the easier and medium batches, the three learned methods have very similar runtimes. On the hardest groups, however, GQSAT develops a much heavier time tail, while ImitSAT and SATformer remain substantially faster. Across this SATCOMP subset, ImitSAT matches SATformer and clearly improves wall-clock time over GQSAT on the most challenging competition formulas. These results show that our evaluation already includes nontrivial SATCOMP instances within our size regime and that, on this subset, ImitSAT attains competitive or best wall-clock performance among the learned branching methods.

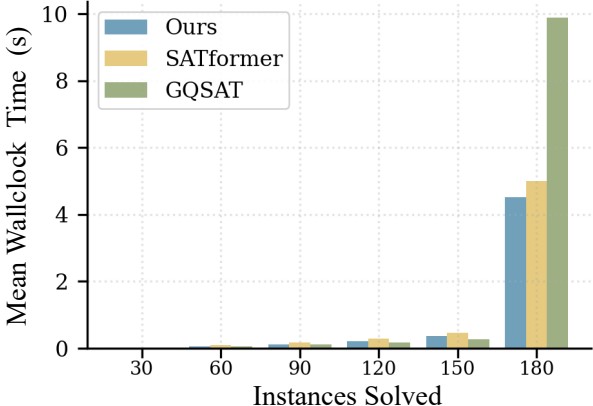

Figure 6: Wall-clock time on the SATCOMP subset. Instances are sorted by runtime and grouped into batches of 30 solved instances; the mean wall-clock time per group is plotted. Across the easier and medium groups, GQSAT, SATformer, and ImitSAT behave similarly, while on the hardest groups (rightmost), GQSAT develops a much heavier time tail and is noticeably slower. In this regime, ImitSAT is the fastest and slightly outperforms SATformer in wall-clock time.

## N    QUERY BUDGET ANALYSIS

In this section, we study how the query budget, the number of times the solver consults the model during search, affects both effectiveness and computational cost. We vary the budget from 1 to 10 calls, and also include an all-calls configuration, to understand how much benefit each additional model query provides.

As shown in Figure 7, the performance gain exhibits clear diminishing returns: the majority of improvement is obtained from the first three queries, while additional calls beyond six provide only marginal benefits. Because each query introduces non-trivial latency, we interpret the reduction in propagations achieved by the model as a benefit that must be weighed against the computational overhead of issuing more queries. Based on this trade-off, we adopt a default budget of 3 queries. This choice is further supported by the wall-clock results in Figure 4, where GQSAT's runtime worsens noticeably as the query budget increases from 3 to 5.

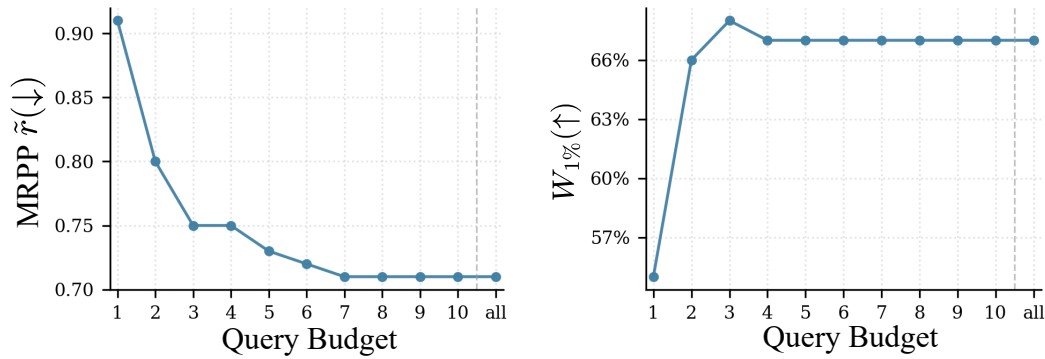

Figure 7: Query budget ablation for ImitSAT on the 5-15 test set. MRPP $\tilde{r}$ (left) and $W_{1\%}$ (right) versus query budget with 1 to 10 calls and the all calls setting, showing that most of the benefit is obtained from the early calls.

## O    WALL-CLOCK TIME COMPARISON WITH MINISAT

In this section, we compare the wall-clock performance of ImitSAT with that of pure MiniSAT and with the other learned baselines. We measure the time (in seconds) required to solve 50 instances for each random 3-SAT range, including all model-inference overheads.

Table 12 reports the results. In the easier 16-30 range, MiniSAT is fastest, as expected, since each instance has few propagations and the cost of a handful of model calls dominates; nevertheless, Imit-SAT remains clearly faster than both GQSAT and SATformer in this regime. For the medium 31-60 range, ImitSAT and MiniSAT have almost identical wall-clock runtimes, and both are substantially faster than the other learned methods. In the 61-100 range, ImitSAT achieves the best wall-clock performance overall, outperforming MiniSAT, GQSAT, and SATformer. Combined with the MRPP $\tilde{r}$ values below one in Tables 1 and 2, these results show that the propagation reductions delivered by ImitSAT translate into net time gains once instances are large and hard enough, while keeping it competitive with MiniSAT and ahead of other learned branching policies even on smaller ranges.

## P    EVALUATION OF NAIVE GNN COMBINATION

To assess whether incorporating embeddings from a trained GNN encoder can improve ImitSAT, we design the following experiment. We adopt the GNN configuration from SATformer and train a message-passing encoder with a self-supervised objective, each true literal–clause edge is treated as a positive sample, and for each positive pair, we construct a negative sample by pairing the same literal with a clause that does not contain it. This yields a GNN encoder that produces embeddings for the CNF formula, which can then be consumed by a downstream autoregressive model.

Table 12: Wall-clock time (seconds) to solve 50 instances on random 3-SAT ranges. For each range, we select the 50 hardest test instances in reverse curriculum order to highlight behaviour on challenging problems. In the easier 16-30 and 31-60 ranges, ImitSAT is the strongest learned method; in the harder 61-100 range, ImitSAT attains the lowest wall-clock runtime overall.

| Method | 16-30 | 31-60 | 61-100 |
|---|---|---|---|
| GQSAT | 1.47 | 3.26 | 7.72 |
| SATformer | 2.56 | 3.38 | 6.50 |
| MiniSAT | **0.53** | **2.54** | 6.44 |
| ImitSAT | 0.85 | 2.70 | **6.11** |

Table 13: Effect of replacing the CNF tokens with GNN embeddings. MRPP $\tilde{r}$ and $W_{1\%}$ on the 5–15 and 16–30 random 3-SAT test sets. The naive GNN-augmented variant underperforms the text-based ImitSAT.

| Metric | Method | 5-15 | 16-30 |
|---|---|---|---|
| MRPP $\tilde{r}$ ($\downarrow$) | use GNN embedding | 1.00 | 0.99 |
| | ImitSAT | **0.75** | **0.83** |
| $W_{1\%}$ ($\uparrow$) | use GNN embedding | 0.46 | 0.50 |
| | ImitSAT | **0.68** | **0.65** |

Given the trained GNN, we replace the tokens between `[CNF]` and `[SEP]` with the corresponding variable embeddings. In other words, we substitute the raw CNF token sequence with the GNN-generated representation in an attempt to inject structural information into the model. We then train the Perceiver-AR model on the 5–15 and 16–30 datasets using this modified input. However, this naive GNN combination does not improve performance. As shown in Table 13, it in fact underperforms the text-only ImitSAT baseline.

In addition, the overall inference cost within the SAT solver must be taken into account. Augmenting the system with a GNN introduces substantial computational overhead during both training and inference, and prior work has shown that GNNs are generally less GPU-friendly and scale poorly on large graphs (Zhang et al., 2025; Han et al., 2024; Liu et al., 2023). Under this naive integration, the GNN-augmented variant is therefore neither more effective nor more efficient than the original text-based ImitSAT.

## Q  TOP-K AND FALLBACK ANALYSIS

This section studies the effectiveness of enforcing decision legality through a Top-K masking strategy. We report the results of the Top-K approach in Table 14, which strictly masks invalid cases and selects the highest-scoring valid decision. On relatively simple in-distribution datasets such as 5–15, 16–30, and 31–60, we observe almost no difference between using or not using Top-K. However, on more challenging out-of-distribution datasets such as JNH and AIM, the Top-K approach leads to degraded performance. One possible explanation is low model confidence in these regimes. When the instance is difficult to model or the solver reaches a hard decision point, the confidence scores become flatter and the selected action is often suboptimal. In these cases, falling back to the original CDCL decision procedure is more reliable than forcing a low confidence model prediction, as shown in Table 14.

## R  INTEGRATION WITH ADVANCED SOLVERS

**CaDiCaL.** We first study how ImitSAT integrates with the modern CDCL solver CaDiCaL (Biere et al., 2024a), using its PySAT (Ignatiev et al., 2018) interface and changing only the branching policy. The preprocessing in CaDiCaL may outright solve a small fraction of instances, so metrics are computed on the remaining instances, but this effect is modest across most benchmarks. Tables 15 and 16 report MRPP $\tilde{r}$ and $W_{1\%}$ under 3 and 5 query budgets, against CaDiCaL default branch-

Table 14: Effect of masking illegal literals with Top-K. MRPP $\tilde{r}$ and $W_{1\%}$ for ImitSAT with and without Top-K on in-distribution (5–15, 16–30, 31–60) and out-of-distribution (JNH, AIM) test sets. Top-K performs similarly on easy instances but degrades on harder benchmarks, where a simple fallback to VSIDS performs better.

| Metric | Method | 5–15 | 16–30 | 31–60 | JNH | AIM |
|--------|--------|------|-------|-------|-----|-----|
| MRPP $\tilde{r}$ ($\downarrow$) | w/ Top-K-3calls | 0.75 | 0.82 | 0.76 | 1.33 | 0.94 |
| | w/ Top-K-5calls | **0.73** | **0.77** | **0.75** | 1.68 | 1.04 |
| | w/o Top-K-3calls | 0.75 | 0.83 | **0.75** | 1.00 | 0.88 |
| | w/o Top-K-5calls | **0.73** | **0.77** | **0.75** | **0.85** | **0.81** |
| $W_{1\%}$ ($\uparrow$) | w/ Top-K-3calls | **0.68** | **0.65** | **0.65** | 0.31 | 0.50 |
| | w/ Top-K-5calls | 0.67 | 0.64 | 0.61 | 0.31 | 0.44 |
| | w/o Top-K-3calls | **0.68** | **0.65** | **0.65** | 0.44 | **0.63** |
| | w/o Top-K-5calls | 0.67 | 0.64 | 0.61 | **0.50** | **0.63** |

Table 15: ImitSAT with CaDiCaL on random 3-SAT test sets. MRPP $\tilde{r}$ ($\downarrow$) and one percent win ratio $W_{1\%}$ ($\uparrow$), where $\tilde{r}$ is the ratio of propagations compared to CaDiCaL with its default branching (baseline = 1.0).

| Metric | Method | 5–15 | 16–30 | 31–60 | 61–100 | 50 | 100 |
|--------|--------|------|-------|-------|--------|-----|-----|
| MRPP $\tilde{r}$ ($\downarrow$) | CaDiCaL baseline | 1.00 | 1.00 | 1.00 | 1.00 | 1.00 | 1.00 |
| | ImitSAT-3calls | **0.73** | 0.74 | 0.73 | **0.63** | 0.73 | **0.66** |
| | ImitSAT-5calls | **0.73** | 0.72 | 0.69 | 0.75 | **0.69** | 0.69 |
| $W_{1\%}$ ($\uparrow$) | CaDiCaL baseline | 0.00 | 0.00 | 0.00 | 0.00 | 0.00 | 0.00 |
| | ImitSAT-3calls | **0.73** | 0.68 | 0.66 | 0.62 | 0.62 | **0.64** |
| | ImitSAT-5calls | 0.71 | 0.68 | 0.66 | 0.62 | 0.62 | 0.61 |

ing. On random 3-SAT test sets, ImitSAT consistently reduces MRPP $\tilde{r}$ and attains strong win ratio $W_{1\%}$, and on structured families it yields clear gains on AIM, PARITY, PHOLE, and especially PRET. These results indicate that ImitSAT transfers effectively to an advanced CDCL solver and improves its performance while only modifying the branching policy.

Table 16: ImitSAT with CaDiCaL on structured SAT families. MRPP $\tilde{r}$ ($\downarrow$) and one percent win ratio $W_{1\%}$ ($\uparrow$), where $\tilde{r}$ is the ratio of propagations compared to CaDiCaL with its default branching (baseline = 1.0).

| Metric | Method | JNH | AIM | PARITY | PHOLE | PRET |
|--------|--------|-----|-----|--------|-------|------|
| MRPP $\tilde{r}$ ($\downarrow$) | CaDiCaL baseline | **1.00** | 1.00 | 1.00 | 1.00 | 1.00 |
| | ImitSAT-3calls | 1.15 | **0.92** | 0.84 | **0.96** | 0.13 |
| | ImitSAT-5calls | 1.46 | 0.95 | **0.66** | 0.99 | **0.10** |
| $W_{1\%}$ ($\uparrow$) | CaDiCaL baseline | 0.00 | 0.00 | 0.00 | 0.00 | 0.00 |
| | ImitSAT-3calls | **0.33** | **0.63** | **0.80** | **0.67** | **1.00** |
| | ImitSAT-5calls | **0.33** | 0.56 | **0.80** | **0.67** | **1.00** |

**Kissat.** Kissat (Biere et al., 2024b) is an aggressively engineered successor of CaDiCaL with heavy pre-/inprocessing, including probing and "lucky" assignments that try speculative assignments before and after preprocessing. On our synthetic 3-SAT benchmarks, these mechanisms often solve the instance without making any CDCL decisions, so the branching callback is rarely invoked. Consistent with this, under the default configuration, the relative propagation statistics of Kissat and Kissat+ImitSAT are almost identical on most random buckets (MRPP $\tilde{r} \approx 1.0$), with noticeable improvements only on the structured datasets. To better isolate the effect of the branching policy, we also consider a search-focused configuration that disables probing, while keeping all other options at their defaults. Table 17 reports results on structured SAT families in this configuration.

Table 17: ImitSAT with Kissat (probing disabled) on structured SAT families. For each family, MRPP $\tilde{r}$ ($\downarrow$) is the median ratio of propagations relative to Kissat without ImitSAT (baseline $= 1.0$), and $W_{1\%}$ ($\uparrow$) is the fraction of instances where ImitSAT reduces propagations by at least 1%.

| Metric | Method | JNH | AIM | PARITY | PHOLE | PRET |
|---|---|---|---|---|---|---|
| MRPP $\tilde{r}$ ($\downarrow$) | Kissat baseline | 1.00 | 1.00 | 1.00 | 1.00 | 1.00 |
| | ImitSAT-3calls | **0.98** | **0.94** | 0.90 | **0.95** | 0.21 |
| | ImitSAT-5calls | 1.00 | 1.03 | **0.81** | 0.96 | **0.06** |
| $W_{1\%}$ ($\uparrow$) | Kissat baseline | 0.00 | 0.00 | 0.00 | 0.00 | 0.00 |
| | ImitSAT-3calls | **0.50** | **0.56** | **0.80** | **0.75** | **1.00** |
| | ImitSAT-5calls | 0.44 | 0.44 | **0.80** | **0.75** | **1.00** |

