# OpenReview forum: "Boolean Satisfiability via Imitation Learning"
_ICLR.cc/2026/Conference — ICLR 2026 Poster_

### Official Review · Reviewer_LTZA · 2025-10-21

**Soundness:** 3
**Presentation:** 3
**Contribution:** 3
**Rating:** 6
**Confidence:** 4

**Summary:**

This paper studies SAT problem. As claimed by the authors, previous learning-based method were limited. In this work, they propose a branching policy for CDCL SAT solvers based on imitation learning. The authors conduct experiments to evaluate the effectiveness of their approach. They claim that their proposed method reduces propagation counts and outperforms previous learning-based method.

**Strengths:**

1. The paper is generally well-written, with clear explanations.

2. The description of the dataset construction method is detailed and reasonable.

3. Experimental results are satisfactory.

**Weaknesses:**

1. What is the SAT solver integrated by SATformer in the experiment? According to the SATformer paper, combining it with CaDiCaL and Kissat can produce performance improvements.

2. MiniSAT, used by ImitSAT, is relatively outdated. How does ImitSAT perform compared to current SOTA solvers (such as Kissat and CaDiCaL)? Can combining ImitSAT with these solvers bring further performance improvements?

**Questions:**

Please reply to my comments listed in ‘Weaknesses’.

**Details Of Ethics Concerns:**

N/A.

---

> ### Author Response · Authors · 2025-11-24
>
> We appreciate the valuable comments, and we have addressed the issue here and in the revised manuscript.
>
> ### **Q1: SATformer and solver integration**
>
> **Response:**
> SATformer is trained as a SAT/UNSAT classifier, and the authors show that the per‑variable “score leading to conflicts” from their model can be used to initialize VSIDS scores. In their paper, this initialization is plugged into CaDiCaL and Kissat, and they report improvements mainly on UNSAT instances.
>
> We started from MiniSAT because it is a small, widely‑used CDCL solver with a simple code path, which lets us isolate the effect of the branching policy without additional engineering choices. It also matches the setup of Graph‑Q‑SAT, another baseline, and makes it easy to interface with a Python model and release a reproducible reference implementation.
>
> ### **Q2: Combining ImitSAT with CaDiCaL and Kissat**
>
> **Response:**
> To address this concern, we have integrated ImitSAT with both CaDiCaL and Kissat and added the detailed per‑dataset results in **Appendix R**.
>
> For **CaDiCaL**, we use the PySAT interface (version 1.9.5) and modify only the branching policy; preprocessing, clause management, and restarts remain unchanged. In these settings, MRPP is defined as the ratio of propagations relative to CaDiCaL with its native branching, so a value of 1.0 corresponds to the unmodified solver and values below 1.0 mean that ImitSAT reduces the number of propagations. With the same small query budgets as in the MiniSAT experiments, ImitSAT drives MRPP below 1.0 and improves the win rate W₁% accordingly. Detailed numbers on random and structured benchmarks are reported in **Tables 15 and 16**.
>
>
> For **Kissat**, strong pre and inprocessing (probing and “lucky” assignments) often solves easier instances before CDCL branching becomes relevant, which limits the impact of any branching policy in the default configuration. To better isolate the effect of branching, we also evaluate a search‑focused configuration with probing disabled, while keeping other options at their defaults. The structured‑family results for this configuration are summarized in **Table 17**. ImitSAT again yields clear reductions in propagations and higher win rates on the harder random and structured benchmarks.
> The table below shows representative results on the PRET structured family. MRPP values below 1 indicate fewer propagations than the default branching of each solver, and W1% is a fraction between 0 and 1.
>
> | Solver + setting            | MRPP (PRET) ↓ | W1% (PRET) ↑ |
> |-----------------------------|---------------|--------------|
> | CaDiCaL baseline            | 1.00          | 0.00         |
> | CaDiCaL + ImitSAT, 3 calls  | 0.13          | 1.00         |
> | CaDiCaL + ImitSAT, 5 calls  | 0.10          | 1.00         |
> | Kissat baseline             | 1.00          | 0.00         |
> | Kissat + ImitSAT, 3 calls   | 0.21          | 1.00         |
> | Kissat + ImitSAT, 5 calls   | 0.06          | 1.00         |
>
>
> These experiments show that ImitSAT is not tied to MiniSAT and can provide further gains when plugged into modern CDCL solvers. We will also release the integration code for both CaDiCaL and Kissat.

---

### Official Review · Reviewer_mcE8 · 2025-10-31

**Soundness:** 3
**Presentation:** 3
**Contribution:** 3
**Rating:** 8
**Confidence:** 3

**Summary:**

The paper introduces ImitSAT, a framework for learning a CDCL decision policy by imitation learning on condensed solver runs. This works by post-processing solver trails to eliminate intermediate conflicts and backtracking steps, which results in compact trails that only contain the key decisions made by the solver. In doing so, the model learns to imitate an improved version of the underlying solver, which can then be used to guide the solver decisions under a certain query budget. The authors compare their method to two other learned hybrid solvers, namely GQSAT and SATformer, in terms of solver-relative performance and wall-clock time on both in-distribution and out-of-distribution benchmarks. Results show that ImitSAT considerably outperforms both baselines on almost all benchmarks under a much lower runtime.

**Strengths:**

1. The paper is clear and easy to follow.
2. The key idea is fairly interesting: distilling a post-processed solver policy into a neural network.
3. The authors compare their method against two somewhat recent baselines.
4. The paper evaluates the method on both in-distribution and out-of-distribution instances.
5. The method outperforms the reported baselines while having a lower runtime.

**Weaknesses:**

1. The authors do not evaluate their method on industrial instances, so it's unclear how well it scales to real-world settings.
2. The attempted query budgets are quite low (up to 5 decisions), so it's unclear how well the method scales with higher budgets.
3. Wall clock time comparisons are missing the pure VSIDS heuristic baseline (i.e., without the model), which would be the ultimate test to see if the proposed method speeds up the CDCL solver.
4. It is unclear in Section 5.1 whether the SATformer and Graph-Q-SAT baseline models were pertrained on their original papers' dataset or on the same random 3-SAT dataset that ImitSAT was trained on. If it's the former, then the comparison in Table 1 is quite unfair since the baselines were trained on a different dataset.
5. Using a text model is arguably not ideal for embedding formulas compared to GNNs as it fails to capture key properties such as invariance to clause permutation and variable renaming. However, I do understand that it might be more convenient to work with text.

**Questions:**

1. Have you tried applying your method to industrial benchmarks like SATCOMP? It does not have to outperform the baseline solver, but it would be good to know where ImitSAT stands in real-world settings.
2. How does the performance look if you only use the model decisions?
3. Have you experimented with higher query budgets than 5? It would be interesting to plot performance over query budget.
4. Equation (10) does not index $F_{DIMACS}$ by $t$, so is it always the original input formula? That is, does it not include the learned clauses? If so, wouldn't that compromise the decision optimality since the model does not see the clauses that were learned during the run?
5. When the model's top answer is illegal, you fall back to the VSIDS heuristic. Why not mask out the illegal variable decisions from the model's output logits/probabilities? In doing so, you could use the model's top legal decision.
6. In Figure 4, can you also include the time of the baseline un-augmented MiniSAT solver?
7. Have you considered bootstrapping the training process by using the model to improve the trails dataset? For example, whenever the model produces a trail with fewer propagations, you could use the produced model trail for subsequent epochs instead of the KeyTrace one.
8. Were the two baselines trained on the same 3-SAT dataset on which ImitSAT was trained, or were they trained on their original papers' datasets?
9. Can you also include the KeyTrace ratios in Tables 1 & 2?
10. To see the time scaling with query budgets, can you include the 5-call ImitSAT to Figure 4, possibly accompanied by higher query budget variants?

---

> ### Author Response · Authors · 2025-11-24
>
> We appreciate the valuable comments, and we have addressed the issue here and in the revised manuscript.
>
> ### **Q1: Industrial / SATCOMP instances.**
>
> **Response:**
> In the revision, we add an experiment on industrial‑style benchmarks from SATCOMP in **Appendix M**. We select instances with at most 8,000 DIMACS and KeyTrace tokens, so that the full competition formula can be passed to the policy without truncation. On this subset, we run MiniSAT, SATformer, Graph‑Q‑SAT, and ImitSAT under the same query budgets as in the main experiments, capping MiniSAT at 10 seconds per instance. As shown in **Figure 6** in Appendix M, the three learned methods behave similarly on the easier and medium batches, but on the hardest groups, Graph‑Q‑SAT develops a much heavier time tail, while ImitSAT and SATformer remain substantially faster. In this regime, ImitSAT slightly outperforms SATformer and achieves **the best wall‑clock performance** overall. These results show that our evaluation already includes nontrivial SATCOMP instances within our size regime and that ImitSAT can improve runtime on real competition formulas under equal compute budgets.
>
>
> ### **Q2: Query budgets and model‑only decisions.**
>
> **Response:**
> In the revision, we add a Query Budget Analysis in **Appendix N**, sweeping budgets from 1 to 10 model calls and also including an “all model calls” setting. **Figure 7** shows that MRPP and W₁% improve as the budget increases, with the largest gains coming from the first three model calls and more moderate improvements thereafter. We view this as a natural trade‑off between reduced propagations and the extra inference cost of additional queries. To keep the integration lightweight, we therefore use 3 model calls as the default budget in the main experiments. The “all model calls” configuration in **Appendix N** illustrates that the policy can in principle drive the solver when every decision is delegated to the model, but our main focus is on small fixed budgets that already capture most of the achievable gains under practical time constraints. These findings indicate that our method scales as expected with higher budgets and that the previously reported results (using up to 5 calls) already capture the major performance trend.
>
>
>
> ### **Q3: Wall‑clock time and pure VSIDS baseline.**
>
> **Response:**
> We agree that comparing against the un‑augmented solver is important. In the revision, we therefore add a direct wall‑clock comparison with pure MiniSAT in **Appendix O**. To keep Figure 4 uncluttered and focused on learned branching methods, the MiniSAT baseline is reported only in the appendix. We measure the time to solve 50 of the hardest test instances in each range (16–30, 31–60, 61–100), including all model‑inference overheads. Runtime (seconds) to solve 50 hardest instances per range:
>
> | Method    | 16–30 | 31–60 | 61–100 |
> |-----------|-------|-------|--------|
> | GQSAT     |  1.47 |  3.26 |  7.72  |
> | SATformer |  2.56 |  3.38 |  6.50  |
> | MiniSAT   |  **0.53** |  **2.54** |  6.44  |
> | ImitSAT   |  0.85 |  2.70 |  **6.11**  |
>
> On the easier 16–30 and 31–60 ranges, MiniSAT is naturally faster because each instance requires few propagations, and the cost of a handful of model calls dominates, yet ImitSAT remains the strongest learned method. On the harder 61–100 range, ImitSAT attains the **lowest wall‑clock runtime overall**, outperforming MiniSAT, GQSAT, and SATformer. This trend mirrors the MRPP values in Tables 1 and 2, where ImitSAT achieves the MRPP of less than 1, indicating that it reduces propagations relative to MiniSAT. Taken together, these results show that the propagation savings delivered by ImitSAT translate into net time gains once instances are large and challenging, while remaining competitive with MiniSAT and ahead of other learned branching policies even on smaller instances.

---

> > ### Author Response · Authors · 2025-11-24
> >
> > ### **Q4: The same training data for SATformer and Graph‑Q‑SAT.**
> >
> >
> > **Response:**
> > We agree that the baselines should also be evaluated under the same data regime. We retrain both SATformer and Graph‑Q‑SAT on our random 3‑SAT dataset, using subsets with comparable size to their original setups. We denote these retrained models by SATformer* and GQSAT*, and we update **Tables 1 and Table 2** accordingly. SATformer* behaves similarly to (and sometimes slightly better or worse than) the original SATformer, while GQSAT* is noticeably worse than GQSAT on most test sets. One possible explanation is that GQSAT* is trained from scratch on our random‑3‑SAT distribution rather than using the transfer curriculum that the original Graph‑Q‑SAT work reports as important for stability and generalization. Taken together, these results indicate that our gains are not due to a more favorable training distribution.
> >
> >
> >
> > For quick reference, the table below summarizes MRPP and W₁% on one representative random bucket (31–60); full per‑dataset results are given in Tables 1 and 2.
> > | Method       | MRPP 31–60 ↓ | W₁% 31–60 ↑ |
> > |-------------|--------------|-------------|
> > | GQSAT-3calls       | 0.89         | 0.54        |
> > | GQSAT*-3calls      | 1.46         | 0.34        |
> > | SATformer   | 0.84         | 0.58        |
> > | SATformer*  | 0.80         | 0.58        |
> > | ImitSAT‑3calls   | 0.75         | 0.65        |
> >
> >
> > ### **Q5: Text model vs GNN encoders.**
> >
> > **Response:**
> > We agree that permutation and renaming invariances are important structural properties of SAT, and that GNN-based encoders naturally capture them. Motivated by this, we follow the SATformer / NeuroSAT design and train a message‑passing GNN with a self‑supervised edge‑prediction objective, then use its final variable embeddings to replace the text \texttt{[CNF]} token sequence. We tuned the GNN depth and message-passing rounds to match the compute budget of our text encoder.
> >
> > Empirically, this variant underperformed our text-only ImitSAT baseline on both MRPP and W₁%. Our interpretation is that although the GNN’s inductive bias is beneficial, it also comes with a cost: the strong structural constraints can limit model expressivity. In contrast, the text model can learn task-specific representations directly from data without imposing these constraints.
> >
> >
> > For quick reference, the table below summarizes the comparison (full details are in **Appendix P**, **Table 13**):
> > | Method             | MRPP 5–15 ↓ | MRPP 16–30 ↓ | W₁% 5–15 ↑ | W₁% 16–30 ↑ |
> > |--------------------|------------:|-------------:|-----------:|------------:|
> > | GNN embeddings     | 1.00        | 0.99         | 0.46       | 0.50        |
> > | Text‑only ImitSAT  | 0.75        | 0.83         | 0.68       | 0.65        |
> >
> > In addition, adding a separate GNN forward pass significantly increases both training and inference cost, and standard message‑passing GNNs are known to scale poorly on GPUs as the graph grows (Zhang et al., 2025; Han et al., 2024; Liu et al., 2023). For these reasons, we keep the simpler text interface in this work. We still view more sophisticated structure‑aware encoders (for example, improved GNNs or permutation‑equivariant transformers) as a promising direction for future work.
> >
> >
> > ---
> >
> > **References:**
> >
> > Guibin Zhang, Xiangguo Sun, Yanwei Yue, Chonghe Jiang, Kun Wang, Tianlong Chen, and Shirui Pan. Graph sparsification via mixture of graphs. In The Thirteenth International Conference on Learning Representations, 2025.
> >
> > Xiaotian Han, Tong Zhao, Yozen Liu, Xia Hu, and Neil Shah. Mlpinit: Embarrassingly simple gnn training acceleration with mlp initialization. In The Eleventh International Conference on Learning Representations, 2024.
> >
> > Zirui Liu, Chen Shengyuan, Kaixiong Zhou, Daochen Zha, Xiao Huang, and Xia Hu. Rsc: accelerate graph neural networks training via randomized sparse computations. In International Conference on Machine Learning, pp. 21951–21968. PMLR, 2023.

---

> > > ### Author Response · Authors · 2025-11-24
> > >
> > > ### **Q6: Learned clauses in Eq. (10).**
> > >
> > > **Response:**
> > > We do not include learned clauses explicitly in the current ImitSAT design. Eq. (10) always conditions on the original input CNF. This is a deliberate choice to keep both the sequence length and the solver–policy interface manageable. Modern CDCL work has repeatedly observed that the learned clause database can grow very large (Audemard & Simon, 2009). If we appended learned clauses as extra tokens, the input length would increase and would soon exceed the model context budget.
> > >
> > > In our implementation, the clause segment is fixed: between [CNF] and [SEP], the model sees only the original DIMACS formula, not the evolving learned database. This gives a hard upper bound on sequence length for every query, independent of how many clauses the solver learns internally. Learned clauses still influence the run indirectly, because all trails and states used for supervision are generated by a full CDCL solver with clause learning. A natural extension, which we leave for future work, is to incorporate a carefully designed compact representation of the learned‑clause database
> > >
> > > ---
> > >
> > > **References:**
> > >
> > > Gilles Audemard and Laurent Simon. Predicting learnt clauses quality in modern sat solvers. In IJCAI, volume 9, pp. 399–404, 2009.

---

> > > > ### Author Response · Authors · 2025-11-24
> > > >
> > > > ### **Q7: Masking illegal literals and fallback.**
> > > >
> > > >
> > > > **Response:**
> > > > We thank the reviewer for this pivot‑thinking suggestion. In early experiments, we implemented exactly this variant: at each model call, we masked out all illegal literals in the logits (already assigned, forced, or otherwise inconsistent) and then selected the highest‑scoring remaining literal. In the revision, we report these results in **Appendix Q: Top‑K and Fallback Analysis**.
> > > >
> > > > On in‑distribution random 3‑SAT sets (5–15, 16–30, 31–60), this masked Top‑K strategy behaves almost identically to our original fallback strategy, where we use the top prediction if it is legal and otherwise revert to the VSIDS decision of the solver. However, on harder out‑of‑distribution families (JNH, AIM), forcing a Top‑K choice consistently degrades performance compared to fallback. The table below summarizes the 3‑call setting on JNH and AIM (full results, including 5 calls and in‑distribution sets, are in Appendix Q, **Table 14**):
> > > > | Method               | MRPP 5–15 ↓ | MRPP AIM↓ | W₁% 5–15 ↑ | W₁% AIM↑ |
> > > > |----------------------|------------:|-----------:|-----------:|----------:|
> > > > | with Top‑K (3 calls) | 0.75        | 0.94       | 0.68       | 0.50      |
> > > > | fallback (3 calls)   | 0.75        | **0.88**       | 0.68       | **0.63**      |
> > > >
> > > >
> > > > Our interpretation is that in these difficult states, the model confidence is low and the score distribution is relatively flat, so forcing a model decision after masking is less effective than falling back to the native heuristic of the solver. In such regimes, the hybrid strategy that follows the model when it is confident but otherwise defers to VSIDS is more robust. In our current design, Top‑K is essentially a passive mask on the logits; a more ambitious direction is to build models that actively encode which literals are legal and how confident they are, so that legality and fallback decisions are part of the learned policy rather than a post hoc filter.
> > > >
> > > >
> > > >
> > > >
> > > > ### **Q8: Bootstrapping training process.**
> > > >
> > > > **Response:**
> > > > Good idea! However, naively replacing KeyTrace supervision with model‑generated trails during training can easily lead to instability, especially in the early stages when the policy is still weak. This kind of on‑policy bootstrapping would also require keeping the solver inside the training loop, which substantially increases the overall computational cost.
> > > >
> > > > A more controlled alternative would be a staged dataset‑bootstrapping scheme, for example, iteratively regenerating improved traces with a stronger policy and retraining on them, but this would convert the pipeline into a multi‑round procedure rather than a single offline supervised pass. We view this as an interesting direction for future work, where techniques from modern large language model training – such as the iterative supervised and reinforcement learning pipeline used in (DeepSeekMath Shao et al.,2024), could be used both for the initial supervised phase and for post-training refinement of the branching policy.
> > > >
> > > > ---
> > > >
> > > > **References:**
> > > >
> > > > Zhihong Shao, Peiyi Wang, Qihao Zhu, Runxin Xu, Junxiao Song, Xiao Bi, Haowei Zhang, Mingchuan Zhang, YK Li, Yang Wu, et al. Deepseekmath: Pushing the limits of mathematical reasoning in open language models. arXiv preprint arXiv:2402.03300, 2024.

---

> > > > > ### Author Response · Authors · 2025-11-24
> > > > >
> > > > > ### **Q9:  Include the KeyTrace ratios in Tables 1 & 2.**
> > > > >
> > > > > **Response:**
> > > > > We included the KeyTrace metrics in both Table 1 and Table 2.
> > > > >
> > > > >
> > > > >
> > > > > ### **Q10: 5-call variants in the time plot.**
> > > > >
> > > > > **Response:**
> > > > > We included both our and Graph-Q-SAT 5 calls in Figure 4.

---

> > > > > > ### Comment · Reviewer_mcE8 · 2025-11-24
> > > > > > **Thank You**
> > > > > >
> > > > > > I thank the authors for their extensive rebuttal and for all the additional evaluations that address my comments.
> > > > > >
> > > > > > Therefore, I will keep my recommendation of acceptance for this paper.

---

> > > > > > > ### Author Response · Authors · 2025-11-24
> > > > > > >
> > > > > > > Thank you very much for taking the time to carefully read our rebuttal and for maintaining your recommendation for acceptance. We truly appreciate your constructive feedback and the opportunity to improve the paper!

---

### Official Review · Reviewer_csYm · 2025-11-01

**Soundness:** 2
**Presentation:** 3
**Contribution:** 2
**Rating:** 4
**Confidence:** 3

**Summary:**

Conflict Driven Clause Learning (CDCL) is the state of the art approach for solving boolean satisfiability (SAT) problems. During CDCL solving, the majority of the solving time is spent on unit propagation, the efficiency of which is heavily influenced by the branching decisions quality.

This paper introduces ImitSAT, which learns to branch in a CDCL solver using imitation learning. To do this, key traces are first extracted by collapsing a CDCL branching search tree into a single sequence of decisions. Then, an autoregressive learner is trained to predict the key traces sequentially through imitation learning. By learning to make better quality branching decisions, ImitSAT is able to cut down solving time. Experiments show that ImitSAT outperforms SATFormer and Graph-Q-SAT.

**Strengths:**

The paper is easy to follow and the idea is simple but effective.

The method is well-motivated by the run-time analysis that propagation is the most time-consuming part of CDCL solving.

The algorithm introduces a budget for querying the model to ensure efficiency.

The approach shows good performance, beating Graph-Q-SAT and SATFormer on structured SAT families outside of training distribution.

**Weaknesses:**

The idea of using imitation learning for branching is not new. For example, it has been successfully applied to guide branch-and-bound for mixed integer programming [1,2].

The algorithm consumes the query budgets greedily, which, given the limited budget could potentially be far from optimal for instances larger than those in Appendix E. Are there simple heuristics that can improve this?

The experiments are on very small sized SAT problems, which provides limited insight to how ImitSAT can be used in practice.

Training is computationally expensive, since CDCL solvers are used to solve a problem entirely to produce a single key trace as a training instance. Therefore, the generalization ability of the method is crucial and calls for a deeper look.


[1] He H, Daumé H, Eisner J. Learning to search in branch and bound algorithms. Advances in neural information processing systems. 2014;27.

[2] Khalil E, Le Bodic P, Song L, Nemhauser G, Dilkina B. Learning to branch in mixed integer programming. InProceedings of the AAAI conference on artificial intelligence 2016 Feb 21 (Vol. 30, No. 1).

**Questions:**

How does the efficiency of ImitSAT maintain as the size of the SAT problem changes? To what sized instance can it generalize?

How is the query budget determined?

---

> ### Author Response · Authors · 2025-11-24
>
> We appreciate the valuable comments. We have addressed the issues here and in the revised manuscript.
>
> ### **Q1: The difference between ImitSAT and branching in Mixed Integer Programming (MIP).**
>
> **Response:**
> Thanks for pointing out the related work by He et al. 2014 and Khalil et al. 2016. We have cited these works in our paper and clarified the connection.
>
> Our work differs from these papers in both the formulation of the branching problem and the methodology of Imitation Learning (IL).
> - The **branching** on SAT CDCL differs from **branching** in MIP branch-and-bound (B&B)/Strong Branching (SB):
>
> First, the MIP B&B explores a monotone tree guided by programming relaxations and global bounds. In contrast, CDCL for SAT is nonmonotone. It allows non‑chronological back jumps and frequent restarts. This yields a DAG-style exploration in which the evolving learned clause database continually reshapes the frontier.
>
> Second, in MIP, there is a local strong‑branching oracle: one can probe relaxations and obtain immediate, node local feedback (bound tightening and pruning). In contrast, CDCL has no comparable per‑node bound oracle; while one can perform lookahead/failed‑literal probing, its effect is mediated by unit propagation, conflicts, and learned clauses and often materializes many steps (and even restarts) later, so feedback is delayed and non‑local.
> These differences, the non-monotone topology, absence of a bound-based local oracle, and delayed, non‑local feedback make “branching” in SAT/CDCL a distinct problem from MIP B&B/strong branching.
> - On the **imitation learning** side, the design and interface also differ:
>
> He et al. (2014) imitate an optimal‑path node‑selection/pruning oracle in programming-based B&B with linear ranking/classification; Khalil et al. (2016) imitate Strong Branching via a ranking SVM on MIP features.
>
> In contrast, our expert is the KeyTrace of surviving CDCL decisions from a full solve run (global, prefix conditioned supervision), our prediction target is the next signed literal (variable and polarity), and our model is a transformer‑based autoregressive generative model. This changes the labels, the task, and the architecture, making our IL formulation distinct from IL in MIP B&B/SB.

---

> > ### Author Response · Authors · 2025-11-24
> >
> > ### **Q2: The greedy query and the budget.**
> >
> > **Response:**
> > We use the budget greedily because we found that early decisions in CDCL shape the CDCL search. Specifically, once the early branch is fixed, later choices have less impact on the trajectory. To elaborate, we show an additional experimental result in Appendix E, where we test on the 5-15 dataset. We found that making a single model call at the first decision is much better than after three VSIDS decisions.
> >
> > To address the concern further, we report additional experiments on larger datasets, both our 100-variable test set and the AIM dataset **(Appendix E, Table 9)**. We compare several schedules under the same budget: (a) 3 calls after 3 VSIDS, (b) 3 calls, each triggered after 3 VSIDS, (c) 3 calls at the beginning (the greedy method). The results below show that schedule (c) achieves both the lowest MRPP and the highest win ratio, indicating that using the query budget as early as possible is already a strong baseline. While more sophisticated budget allocation methods may yield gains, designing them is a separate research question that requires further investigation based on specific SAT properties.
> >
> > | Schedule                      | MRPP 100 ↓ | MRPP AIM ↓ | W₁% 100 ↑ | W₁% AIM ↑ |
> > |------------------------------|------------|------------|-----------|-----------|
> > | 3 calls after 3 VSIDS        | 0.87       | 1.02       | 0.57      | 0.13      |
> > | 3 calls, 1 per 3 VSIDS       | 0.95       | 1.09       | 0.52      | 0.19      |
> > | 3 calls at first decision    | **0.76**   | **0.88**   | **0.60**  | **0.63**  |
> >
> > **For the query budget**, we treat it as a trade-off between reduced propagation and increased inference cost. **Tables 1 and 2, and Figure 7 in Appendix N**, show results for different query budgets on the MRPP and win ratio. We found that the gain diminishes as the query budget increases. **Figure 4** includes the 5 model calls, both ours and Graph-Q-SAT, and shows that both methods have worse wall-clock time. For quick reference, the table below shows the effect of 3 versus 5 calls on the 5–15 test set (full per‑dataset numbers are in Tables 1 and 2, runtimes in Figure 4):
> > | Method  | Budget   | MRPP (5–15) ↓ | W₁% (5–15) ↑ | Wall‑clock for 5k instances (s) |
> > |---------|----------|---------------|--------------|----------------------------------|
> > | ImitSAT | 3 calls  | 0.75          | 0.68         | **≈ 40**                             |
> > | ImitSAT | 5 calls  | **0.73**          | **0.67**         | ≈ 70                             |
> > | GQSAT   | 3 calls  | 1.00          | 0.45         | ≈ 95                             |
> > | GQSAT   | 5 calls  | 1.00          | 0.46         | ≈ 125                            |
> >
> >
> > Therefore, we suggest using 3 model calls as the default query budget.

---

> > > ### Author Response · Authors · 2025-11-24
> > >
> > > ### **Q3: The instance size, model efficiency, and generalization.**
> > >
> > > **Response:**
> > > Our experiments focus on SAT instances whose DIMACS and KeyTrace encodings fit within our current 8k‑token context window. This is a similar or larger regime than in prior learning‑based branching work, such as SATformer and Graph-Q-SAT. In the revision, we also add an experiment on industrial‑style benchmarks from SATCOMP in **Appendix M**. We selected instances whose DIMACS and KeyTrace segments both fit into this context, so the full competition formula can be passed to the policy without truncation, and we ran MiniSAT, SATformer, Graph‑Q‑SAT, and ImitSAT under the same query budgets, capping MiniSAT at 10 seconds per instance. As shown in **Figure 6** in Appendix M, the three learned methods behave similarly on the easier and medium batches, but on the hardest groups Graph‑Q‑SAT develops a much heavier time tail, while ImitSAT and SATformer remain substantially faster; in this regime, ImitSAT slightly outperforms SATformer and attains the best wall‑clock performance. These results show that our evaluation already includes real SATCOMP instances within our size regime and that ImitSAT can improve runtime on industrial benchmarks under equal compute budgets.
> > >
> > >
> > > Importantly, our results demonstrate robust generalization beyond the training distribution. Although the model is trained only on simple random 3-SAT instances, it produces measurable improvements on UNSAT benchmarks, non-3-SAT formulas, and several structured families. This out-of-distribution generalization provides evidence that the learned policy captures non-trivial CDCL decision patterns and can transfer to more realistic settings.
> > >
> > > We agree that many industrial SAT instances can be larger, and our current experimental scale is partly limited by the computational resources available to our lab. However, the design of ImitSAT is inherently scalable. Each model query requires only the next CDCL decision, and the computational cost of the model is $O(N)$ per query, in contrast to the $O(N^{2})$ complexity of standard transformer decoders. With a fixed query budget, the overhead from our learned branching policy therefore grows linearly with the problem size. This makes it straightforward to extend our method to larger instances when additional compute resources and a larger context budget are available.
> > >
> > >
> > >
> > > ### **Q4: The training computation cost and the generalization ability.**
> > >
> > >
> > > **Response:**
> > > We agree that generalization and scalability are crucial for practical impact, and our pipeline is designed with both goals in mind.
> > >
> > > **Data generation.** Each training instance is solved once by a standard CDCL solver to obtain a full trail, which we then compress into a KeyTrace. This step is offline and highly parallelizable (one instance per CPU core), and a KeyTrace with $D$ decisions yields $D$ supervised examples from its prefixes.
> > >
> > > **Model and training.** Our policy is a transformer-based autoregressive Perceiver‑AR model with ~160M parameters, trained on 4 V100 GPUs. Training is a standard supervised sequence‑model loop with no solver in the training loop, so it behaves like a normal sequence model. Each inference call has $O(N)$ cost in the input length $N$.
> > >
> > > **Generalization.** Although the model is trained only on random 3‑SAT instances, it transfers in our experiments to UNSAT benchmarks, non‑3‑SAT formulas, and structured families. The table below summarizes ImitSAT with 3 calls on several structured families (Table 2 in the paper); MRPP values below 1 indicate fewer propagations than the MiniSAT baseline, and W₁% is the fraction of instances with at least a 1% propagation reduction:
> > >
> > >
> > > | Family | MRPP (ImitSAT, 3 calls) ↓ | W₁% (ImitSAT, 3 calls) ↑ |
> > > |--------|---------------------------|---------------------------|
> > > | JNH    | 1.00                      | 0.44                      |
> > > | AIM    | 0.88                      | 0.63                      |
> > > | PARITY | 0.30                      | 0.80                      |
> > > | PHOLE  | 1.00                      | 0.50                      |
> > > | PRET   | 0.42                      | 1.00                      |
> > >
> > >
> > > This out‑of‑distribution behavior shows that, despite being trained only on random 3‑SAT, the learned policy improves propagation metrics on multiple structured benchmarks, indicating that it captures non‑trivial CDCL decision patterns rather than overfitting to the training distribution.

---

### Meta-Review · Area_Chair_fLxj · 2025-12-12

**Summary:**

The paper proposes an imitation learning approach for SAT.  The reviewers expressed the following concerns:

1. Imitation learning for branching is not new
2. The experiments are on very small sized SAT problems
3. Training is computationally expensive
4. No evaluation on industrial instances
5. The attempted query budgets are quite low (up to 5 decisions)
6. Wall clock time comparisons are missing the pure VSIDS heuristic baseline
7. No comparison to current SOTA solvers (such as Kissat and CaDiCaL)

**Reviewer Concerns:**

The authors addressed all the concerns of the reviewers.  While there has been previous work that explored imitation learning in branch and bound as well as mixed integer programming, the proposed technique is novel since preparation of the training set is different.  The paper explains how to collapse traces into key traces that contain only the relevant events.  The authors included additional experiments with industrial instances, larger problems and comparisons with SOTA solvers such as Kissat and CaDiCaL.  The authors also reported the training time and included a study of the impact of the query budgets.

**Reviewer Scores:**

I expect reviewer csYm to raise its score and recommend acceptance since the rebuttal addressed all of the concerns.

I expect reviewers mcE8 and LTZA to keep their positive scores since their concerns were not critical and the authors addressed them well anyway.

Overall, this is very nice work that advances the state of the art for learning based techniques in SAT.  The approach is simple and practical.  The empirical results demonstrate well the benefits of the approach.

---

### Decision · Program_Chairs · 2026-01-26

Accept (Poster)